# 🔍 Q-MIRROR: UNLOCKING THE MULTI-MODAL POTENTIAL OF SCIENTIFIC TEXT-ONLY QA PAIRS

## ABSTRACT

High-quality, multi-modal benchmarks are crucial for advancing scientific reasoning in large models yet their manual creation is costly and unscalable. To address this bottleneck, we explore the potential for transforming Text-Only QA Pairs (TQAs) into high-quality Multi-Modal QA Pairs (MMQAs), which include three parts: 1) **Task Definition & Evaluation Rubric**: We develop a TQA-to-MMQA framework and establish a comprehensive, multi-dimensional MMQA quality rubric that provides principles for the transformation. 2) **Benchmark Construction**: Then we construct two extensive benchmarks to rigorously evaluate state-of-the-art generation & understanding models on the distinct tasks of MMQA generation & MMQA quality evaluation. 3) **Preliminary Solution**: We develop an agentic system *(Q-Mirror)*, which operationalizes our framework by integrating MMQA generation and evaluation into a closed loop for iterative refinement. Our experiments show that while state-of-the-art models can generate MMQAs, their outputs still leave substantial gaps, underscoring the need for reliable evaluation. We further demonstrate that top-tier understanding models align closely with human judgment in MMQA quality assessment. Leveraging both insights, the Q-Mirror agent raises average scores from 78.90 to 85.22 and pass rates from 72% to 95%, offering a practical path to large-scale scientific benchmarks.

## 1 INTRODUCTION

High-quality scientific data is the core to the benchmark of large scientific models (Bai et al., 2025; Hu et al., 2025; Qin et al., 2025). Since text-only data are relatively easy to collect and standardize, the community has built extensive banks of text-only QA pairs. However, real-world scientific problem solving is inherently multi-modal, often requiring the integration of visual diagrams, formulas, charts, and experimental setups. Thus, the demand for multi-modal scientific data is both clear and urgent. In response, several efforts have attempted to curate multi-modal benchmarks, yet the collection of such data is far more challenging, which requires higher annotation costs, richer domain expertise, and complex formatting. It has resulted in a pronounced imbalance: text-only resources greatly outnumber multi-modal ones. This observation motivates our guiding question:

*Why not transform existing text-only QA pairs into multi-modal ones?*

Interestingly, many existing text-only resources already contain implicit multi-modal cues (Wang et al., 2024a; Zhang et al., 2025a; Wang et al., 2025d). Geometry problems often reference diagram, physics questions describe experimental setups, and chemistry tasks mention molecular structures or reaction schemes (Cho et al., 2025; Ning et al., 2023; Küchemann et al., 2025; Yin et al., 2024). For instance, a physics problem asking to calculate the trajectory of a projectile, while described textually, implicitly calls for a diagram illustrating the initial velocity, angle, and gravitational force. These latent visual or structural elements, though expressed textually, could be surfaced and enriched to create multi-modal QA pairs (MMQAs). If such transformations were possible at scale, they would unlock the value of vast text-only banks without incurring the full cost of new multi-modal data collection. This perspective reframes the challenge: rather than treating text-only and multi-modal resources as disjoint, we explore how to systematically convert one into the other to accelerate multi-modal benchmark construction and, ultimately, advance scientific reasoning in large models.

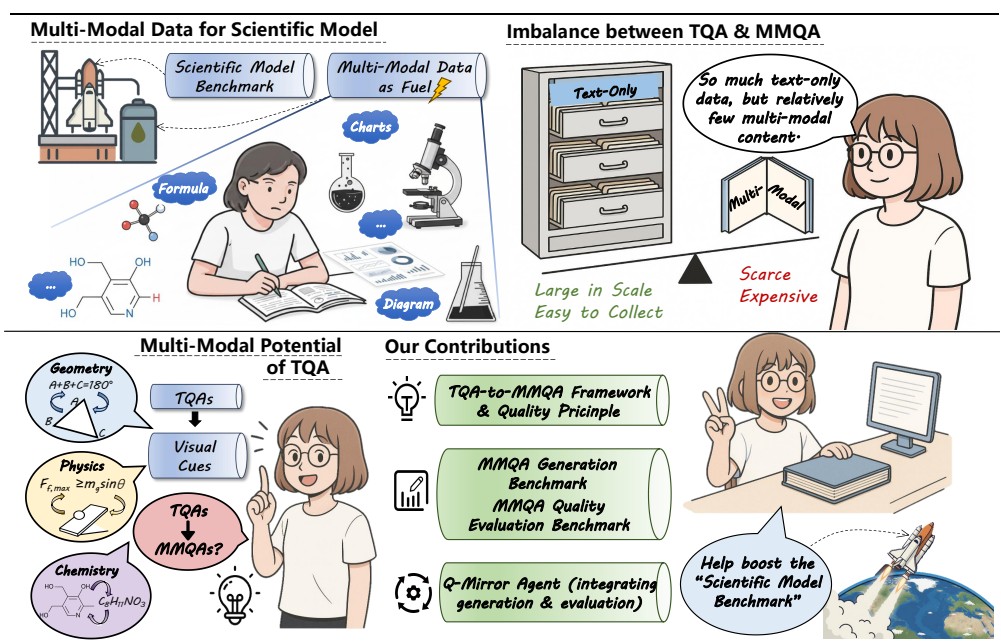

Figure 1: Overview of the motivation and key contributions, which illustrate: 1) the need for multi-modal data in scientific benchmark to advance model development, 2) the imbalance between text-only and multi-modal resources, 3) the latent potential of TQAs for multi-modal transformation, and 4) our contributions, including the transformation framework and quality principles, MMQA generation and evaluation benchmarks, and the Q-Mirror Agent for improving MMQA quality.

Despite this potential, there is still no systematic framework for converting text-only scientific questions into multi-modal formats. Existing datasets treat text-only and multi-modal resources as largely disjoint, and current augmentation methods are limited to narrow tasks or small scales (Wang et al., 2025e; Liu et al., 2023). Consequently, the vast repositories of TQAs remain underutilized, and the lack of scalable transformation tools constrains the development and evaluation of large scientific models. The rapid advancements in large language models (LLMs) and, more recently, large multi-modal models (LMMs), have opened unprecedented opportunities for complex content generation and understanding, making this systematic transformation more feasible than ever before.

Therefore, we explore the framework designed to transform TQAs into high-quality MMQAs. First, we establish a principled quality rubric to rigorously define what constitutes a successful transformation. Second, using this rubric as a foundation, we conduct two extensive benchmarks that evaluate state-of-the-art models on MMQA generation and evaluation, mapping out the current landscape of capabilities. Finally, armed with insights from these benchmarks, we develop a novel agentic system (Q-Mirror) that operationalizes our framework, orchestrating generation and judgment in a closed-loop process to autonomously produce high-quality, refined MMQAs. As illustrated in Figure 1, these elements collectively motivate and shape our framework. Building on this foundation, our contributions can be summarized as follows:

- **A Systematic TQA-to-MMQA Framework and Quality Rubric.** We propose the first systematic framework for the TQA-to-MMQA transformation, grounded in a comprehensive, multi-dimensional quality rubric. This rubric establishes principled criteria for high-quality MMQAs and serves as the foundation for entire study.

- **Benchmarks on MMQA Generation & MMQA Quality Evaluation.** Grounded in our rubric, we construct and release two extensive benchmarks that evaluate state-of-the-art models on distinct tasks of 1) MMQA generation from TQAs, and 2) MMQA quality evaluation. These benchmarks reveal the capabilities of the current LLMs on complex generation and understanding.

- **An Agentic System for MMQA Generation & Refinement.** Building upon the benchmark findings, we develop **Q-Mirror**, a novel agentic system that integrates MMQA generation and judgment. It operates in a closed-loop, iterative refinement process to autonomously convert TQAs into high-quality MMQAs at scale, providing a practical solution to the data scarcity problem.

Table 1: Comparative Analysis of Scientific Benchmarks. This table contrasts key attributes of existing benchmarks to highlight **the scarcity of multi-modal resources and their high construction costs**, which motivates our work. **Legend:** *Subjects*: ■ Math, ■ Physics, ■ Chemistry, etc. *Eval. Methods*: ■ rule- / ■ model- / ■ answer-level based. *Difficulty*: ■ Knowledge Recall, ■ Concept Application, ■ Cognitive Reasoning. *#Multi-Modal Data Construction*: ■ Manual / ■ Automatic annotation.

| Benchmark | Subjects | Multi-modal | Language | Size | #Answer | Eval. | Difficulty | #Data. |
|---|---|---|---|---|---|---|---|---|
| C-Eval (Huang et al., 2023) | | × | ZH | 13948 | 1 | ■ / ■ | | - |
| MATH (Hendrycks et al., 2021) | | × | EN | 12500 | 1 | ■ / ■ | | - |
| SciEval (Sun et al., 2024) | | × | EN | 15901 | 4 | ■ / ■ | | - |
| SuperGPQA (Team et al., 2025) | | × | EN | 26529 | 3 | ■ / ■ | | - |
| MMLU (Hendrycks et al., 2020) | | × | EN | 15908 | 1 | ■ / ■ | | - |
| CMMLU (Li et al., 2024a) | | × | ZH | 11528 | 1 | ■ / ■ | | - |
| MMLU-Pro (Wang et al., 2024c) | | × | EN | 12032 | 1 | ■ / ■ | | - |
| TriviaQA (Joshi et al., 2017) | | × | EN | 95956 | 3 | ■ / ■ | - | - |
| EESE (Wang et al., 2025c) | | × | EN | 100000+ | 5 | ■ / ■ | | - |
| AI2D (Kembhavi et al., 2016) | | ✓ | EN | 15000+ | 1 | ■ / ■ / ■ | - | ■ |
| ScienceQA (Lu et al., 2022) | | ✓ | EN | 21208 | 1 | ■ / ■ | | ■ |
| MathVista (Lu et al., 2023) | | ✓ | EN | 6141 | 4 | ■ / ■ / ■ | | ■ |
| MMMU (Yue et al., 2024) | | ✓ | EN | 11500 | 2 | ■ / ■ | | ■ |
| ChemVLM (Li et al., 2025) | | ✓ | EN&ZH | 4700+ | 4 | ■ / ■ | - | ■ |
| HLE (Phan et al., 2025) | | ✓ | EN | 3000 | 4 | ■ | | ■ |

## 2 RELATED WORK

**Scientific Benchmarks.** The advancement of scientific LLMs is fundamentally intertwined with the availability of high-quality data resources. Classical text-only resources such as TriviaQA (Joshi et al., 2017), MMLU (Hendrycks et al., 2020), C-Eval (Huang et al., 2023), and GPQA (Rein et al., 2024) cover diverse domains and reasoning levels at scale. In contrast, the recent emergence of LMMs has stimulated benchmarks like AI2D (Kembhavi et al., 2016), ScienceQA (Lu et al., 2022), MathVista (Lu et al., 2023), and MMMU (Yue et al., 2024), which emphasize visual–language integration and cross-disciplinary coverage. However, as shown in Table 1, the number of instances in text-only benchmarks far exceeds those of multi-modal ones (Li et al., 2024b; Zhang et al., 2025b; Wang et al., 2025b). This imbalance highlights a structural gap: *the rapid progress of scientific LMMs stands in tension with the limited scale and availability of multi-modal resources.*

**Multi-Modal Data Construction.** Current multi-modal scientific resources are primarily *manually curated*. For example, AI2D (Kembhavi et al., 2016) is constructed with expert-labeled diagrams and dense annotations, ScienceQA (Lu et al., 2022) integrates web-harvested questions with human validation, and MathVista (Lu et al., 2023) and MMMU (Yue et al., 2024) merge domain-specific datasets through expert-driven curation. Although manual construction guarantees quality, it is inherently expensive, time-consuming, and difficult to scale Villalobos et al. (2022); Wang et al. (2025a). While a few efforts explore automatic generation, such as ChemVLM (Li et al., 2025), they remain limited in coverage and domain generality. These limitations motivate a new paradigm: *systematically transforming abundant text-only resources into multi-modal ones.*

**LLMs as Judges.** Parallel to benchmark construction, recent studies demonstrate the promise of LLMs as not only problem solvers but also *judges* for evaluation (Zheng et al., 2023). They can deliver rubric-grounded absolute scores with natural language explanations, or conduct pairwise preference ranking (Wadhwa et al., 2025; Liu et al., 2025). Their reliability is often stress-tested using counterfactual or perturbation-based methods, with consistency against human ratings quantified by correlation and calibration metrics (Tan et al., 2025; Chan et al., 2023). Building on this line of work, we introduce an *agentic system* that integrates both generation and evaluation, allowing LMMs to iteratively refine multi-modal scientific QA pairs. This strategy extends the 'LLM-as-Judge' paradigm and unlocks the latent potential of large-scale text-only corpora.

## 3 TASK DEFINITION & EVALUATION RUBRIC

To systematically realize TQAs into high-quality MMQAs, we first introduce a general framework for TQA-to-MMQA transformation (§3.1). Next, we define a rubric that establishes principled criteria for high-quality MMQAs (§3.2).

### 3.1 FRAMEWORK FOR TQA-TO-MMQA TRANSFORMATION

To unlock the latent multi-modal potential within existing text-based resources, we first establish a scalable and well-controlled process for systematically converting TQAs into MMQAs.

Table 2: Brief description for the MMQA quality rubric, which includes three principles: Information Consistency (IC), Cross Modal Integration (CM), and Standalone Quality (QT).

| Principle | Metric | Notation | Description |
|---|---|---|---|
| **IC** | IL / Information Loss | $p_-$ | *If critical information from the original TQA is lost.* |
| | IA / Information Addition | $p_+$ | *If critical information from the original TQA is added.* |
| **CM** | SI / Solvability with Image | $p_s(I)$ | *If the question can be solved from the image alone.* |
| | SQ / Solvability with Question | $p_s(T)$ | *If the question can be solved from the text alone.* |
| | RE / Redundancy-Synergy | $f_s$ | Evaluates the degree of information overlap. |
| **QT** | NE / Natural Expression | $p_{\text{nat}}$ | *If the text is linguistically fluent and coherent.* |
| | TQ / Technical Quality | $p_{\text{tech}}$ | *If the image is technically correct and artifact-free.* |
| | AQ / Aesthetic Quality | $p_{\text{aes}}$ | *If the image is visually clear and appealing.* |
| | SC / Semantical Clarity | $p_{\text{sem}}$ | *If the image is plausible and scientifically sound.* |

*1) The Modal Conversion Stage.* For the given TQA, we use an LMM to first identify the parts that can be transformed into visual information. The LMM then removes these segments from the text and replaces them with an image placeholder, ensuring that the resulting question becomes visually dependent. Meanwhile, the LMM generates a detailed visual description based on the removed content, which serves as input for the text-to-image (T2I) model (*detailed prompts in Appendix D*).

*2) The Image Generation Stage.* A T2I model is utilized to synthesize the image using the detailed description from the previous stage. To ensure semantic alignment and stylistic coherence, we emphasize that the description generator (LMM) and the image generator (T2I model) should ideally belong to the **same model family** (e.g., *GPT-4.1-2025-04-14* (OpenAI, 2025a) and *GPT-Image-1* (OpenAI, 2025b)). Such family-level consistency often facilitates a closer coupling between textual semantics and visual synthesis, helping to reduce modal gaps and improve the generation quality.

### 3.2 Quality Principles for MMQA Transformation

A good MMQA transformation should satisfy three key criteria. First, it must maintain *information consistency*, ensuring that the essential meaning of the original TQA is preserved without losing or introducing critical details. Second, it should achieve *cross-modal integration*, where the question and image provide complementary information such that neither modal alone is sufficient for solving the problem, and true multi-modal dependence is enforced. Third, it must guarantee high *standalone quality*, with the text remaining fluent and natural, and the image being technically correct, visually clear, and scientifically sound. Therefore, we establish a multi-dimensional rubric grounded in three principles: information consistency, cross-modal integration, and standalone quality (*Table 2*).

#### 3.2.1 Preliminaries

Let $Q_s$ be an original TQA and $Q_m = (T, I)$ be its transformed counterpart. The evaluation relies on a set of boolean predicates $\mathbb{I}[\cdot]$, which return true (1) or false (0). The composite scores for each principle are defined below, scaled from 0 to 100. To ensure reproducibility, each predicate is evaluated against specific operational criteria (*details can be referred to in Appendix C*).

#### 3.2.2 Evaluation Principles

*Principle 1: Information Consistency (IC).* This principle measures semantic fidelity between the source TQA $Q_s$ and its transformed MMQA $Q_m$. We define a predicate set $\mathcal{P}_{\text{IC}} = \{p_-, p_+\}$, where $p_-$ detects missing critical information and $p_+$ detects spurious additions. The IC score is the complement of their violation rate:

$$\text{IC}(Q_m, Q_s) = 1 - \frac{1}{|\mathcal{P}_{\text{IC}}|} \sum_{p \in \mathcal{P}_{\text{IC}}} \mathbb{I}[p(Q_m, Q_s)], \tag{1}$$

*Principle 2: Cross-Modal Integration (CM).* This principle evaluates whether text and image contribute complementary, indispensable information. We test single-modal solvability using $p_s(T)$ and $p_s(I)$, and assess synergy via $f_s(T, I) \in \{Complete, Partial, None\}$. Let $\beta$ denote the weight mapping of overlap categories. The CM score is defined as:

$$\text{CM}(Q_m) = \frac{1}{3}\Big((1 - \mathbb{I}[p_s(T)]) + (1 - \mathbb{I}[p_s(I)]) + \beta[f_s(T, I)]\Big), \tag{2}$$

where we set $\beta[Partial] = 0.75$, $\beta[None] = 0.25$, and $\beta[Complete] = 0$.

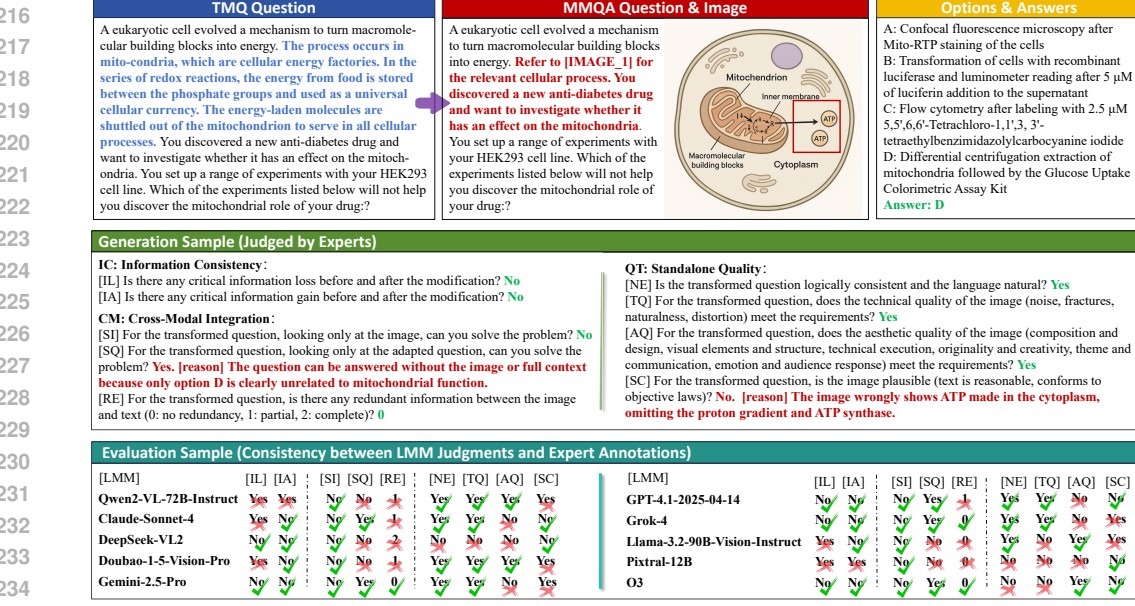

**Generation Sample (Judged by Experts)**

**IC: Information Consistency**:
[IL] Is there any critical information loss before and after the modification? **No**
[IA] Is there any critical information gain before and after the modification? **No**

**CM: Cross-Modal Integration**:
[SI] For the transformed question, looking only at the image, can you solve the problem? **No**
[SQ] For the transformed question, looking only at the adapted question, can you solve the problem? **Yes. [reason] The question can be answered without the image or full context because only option D is clearly unrelated to mitochondrial function.**
[RE] For the transformed question, is there any redundant information between the image and text (0: no redundancy, 1: partial, 2: complete)? **0**

**QT: Standalone Quality**:
[NE] Is the transformed question logically consistent and the language natural? **Yes**
[TQ] For the transformed question, does the technical quality of the image (noise, fractures, naturalness, distortion) meet the requirements? **Yes**
[AQ] For the transformed question, does the aesthetic quality of the image (composition and design, visual elements and structure, technical execution, originality and creativity, theme and communication, emotion and audience response) meet the requirements? **Yes**
[SC] For the transformed question, is the image plausible (text is reasonable, conforms to objective laws)? **No. [reason] The image wrongly shows ATP made in the cytoplasm, omitting the proton gradient and ATP synthase.**

**Evaluation Sample (Consistency between LMM Judgments and Expert Annotations)**

| [LMM] | [IL] | [IA] | [SI] | [SQ] | [RE] | [NE] | [TQ] | [AQ] | [SC] |
|---|---|---|---|---|---|---|---|---|---|
| Qwen2-VL-72B-Instruct | Yes ✓ | Yes ✗ | No ✓ | No ✓ | -1 ✗ | Yes ✓ | Yes ✓ | Yes ✓ | Yes ✗ |
| Claude-Sonnet-4 | Yes ✓ | No ✓ | No ✓ | Yes ✓ | -1 ✗ | Yes ✓ | No ✗ | No ✓ | No ✓ |
| DeepSeek-VL2 | No ✗ | No ✓ | No ✓ | No ✗ | -2 ✗ | No ✗ | No ✗ | No ✗ | No ✓ |
| Doubao-1-5-Vision-Pro | Yes ✓ | No ✓ | No ✓ | Yes ✓ | -1 ✗ | Yes ✓ | Yes ✓ | Yes ✓ | Yes ✗ |
| Gemini-2.5-Pro | No ✗ | No ✓ | No ✓ | Yes ✓ | 0 ✓ | Yes ✓ | Yes ✓ | No ✗ | Yes ✗ |

| [LMM] | [IL] | [IA] | [SI] | [SQ] | [RE] | [NE] | [TQ] | [AQ] | [SC] |
|---|---|---|---|---|---|---|---|---|---|
| GPT-4.1-2025-04-14 | No ✗ | No ✓ | No ✓ | Yes ✓ | -1 ✗ | Yes ✓ | Yes ✓ | No ✗ | No ✓ |
| Grok-4 | No ✗ | No ✓ | No ✓ | Yes ✓ | 0 ✓ | Yes ✓ | Yes ✓ | No ✗ | Yes ✗ |
| Llama-3.2-90B-Vision-Instruct | Yes ✓ | No ✓ | No ✓ | No ✗ | 0 ✓ | Yes ✓ | No ✗ | Yes ✓ | Yes ✗ |
| Pixtral-12B | Yes ✓ | Yes ✗ | No ✓ | No ✗ | 0 ✗ | No ✗ | No ✗ | No ✗ | No ✓ |
| O3 | No ✗ | No ✓ | No ✓ | Yes ✓ | 0 ✓ | No ✗ | No ✗ | Yes ✓ | No ✓ |

Figure 2: An illustration of the TQA-to-MMQA transformation, expert annotation, and LMM judge evaluation, including: 1) the full conversion of a TQA into an MMQA, 2) the expert annotation sample for the corresponding MMQA generation case, based on the proposed quality rubric, and 3) the evaluation results of LMMs with their correctness indicated against expert annotations.

*Principle 3: Standalone Quality (QT).* This principle assesses the intrinsic quality of each modal. We define $\mathcal{P}_{\text{QT}} = \{p_{\text{nat}}(T), p_{\text{tech}}(I), p_{\text{aes}}(I), p_{\text{sem}}(I)\}$, covering textual fluency and visual adequacy. The QT score is the average success rate:

$$\text{QT}(Q_m) = \frac{1}{|\mathcal{P}_{\text{QT}}|} \sum_{p \in \mathcal{P}_{\text{QT}}} \mathbb{I}[p(Q_m)]. \tag{3}$$

*Final Aggregated Score (AVG).* The overall score is a weighted sum of the three principles:

$$\text{AVG} = \alpha_{\text{IC}} \cdot \text{IC}(Q_m, Q_s) + \alpha_{\text{CM}} \cdot \text{CM}(Q_m) + \alpha_{\text{QT}} \cdot \text{QT}(Q_m), \tag{4}$$

with $\alpha_{\text{IC}} + \alpha_{\text{CM}} + \alpha_{\text{QT}} = 1$. We set $\alpha_{\text{IC}} = 0.3$, $\alpha_{\text{CM}} = 0.3$, and $\alpha_{\text{QT}} = 0.4$, reflecting that standalone quality is a prerequisite for usability.

## 4 BENCHMARK CONSTRUCTION

In this section, we first introduce the TQA collection (§4.1). Then, we construct a benchmark for TQA-to-MMQA generation (§4.2), and finally, we validate the capabilities of popular LMMs as automated judges for MMQA quality evaluation (§4.3). A corresponding case is shown in Figure 2, which illustrates the TQA-to-MMQA transformation, expert annotation based on the proposed rubric, and the comparison of LMM judgments with human annotations.

### 4.1 TQA COLLECTION

We begin by sampling candidate TQAs from multiple authoritative scientific benchmarks (Rein et al., 2024; Wang et al., 2025c) to ensure diversity and disciplinary breadth. An LMM is then employed to automatically detect which questions are suitable for transformation into multi-modal form, discarding those that lack clear visualizable elements. Next, human experts review the remaining questions to verify their scientific value, refine their formulations, and guarantee balanced coverage across 22 scientific disciplines. Finally, the curated pool is partitioned into two subsets according to difficulty, determined jointly by dataset-provided difficulty labels and expert judgment: **Q-Mirror-Expert (310 questions)** focuses on frontier or interdisciplinary concepts that often lack standard visual representations, while **Q-Mirror-Grad (130 questions)** targets well-established graduate-level knowledge requiring precise visual reproduction of structured scientific information. Further details on TQA Collection are provided in Appendix B.

---

**Algorithm 1** Q-Mirror: Iterative Refinement Workflow

---

1: **Input:** Original TQA $\mathcal{Q}_s$, Quality Threshold $\tau$, Max Iterations $N$, Candidates per Iteration $K$.
2: **Output:** High-quality MMQA $\mathcal{Q}_m^*$.
3: Initialize $feedback \leftarrow$ null, best_score $\leftarrow -\infty$, $\mathcal{Q}_m^* \leftarrow$ null
4: **for** $i \leftarrow 1$ to $N$ **do**
5:     Candidates $\leftarrow$ **Planner.GenerateCandidates**($\mathcal{Q}_s, feedback, K$)
6:     best_cand, best_score_curr, judgments $\leftarrow$ **Evaluation.EvaluateBatch**(Candidates)
7:     **if** best_score_curr > best_score **then**
8:         best_score $\leftarrow$ best_score_curr
9:         $\mathcal{Q}_m^* \leftarrow$ best_cand
10:     **end if**
11:     **if** best_score $\geq \tau$ **then break**
12:     **end if**
13:     $feedback \leftarrow$ **Controller.GenerateFeedback**(judgments)
14: **end for**
15: **return** $\mathcal{Q}_m^*$

---

## 4.2 MMQA GENERATION BENCHMARK

To establish a gold-standard benchmark for MMQA generation, we institute a rigorous human evaluation protocol grounded in our core principles (§3.2). We assemble a panel of 50 domain experts, primarily researchers in STEM fields, who experience an intensive training and calibration process. Each model-generated MMQA is independently assessed by at least two experts using the fine-grained rubric that is based on our principles. Experts assign multi-dimensional scores and provide textual justifications for any identified flaws. This effort culminates in the MMQA generation benchmark, a dataset enriched with expert-verified quality labels. A complete formalization of our rubric, including detailed guidelines and examples provided to annotators, is available in Appendix D.

## 4.3 MMQA EVALUATION BENCHMARK

While human annotation provides the gold standard, its high cost and low scalability are critical bottlenecks for iterative development. To overcome this, we systematically evaluate the viability of state-of-the-art LMMs as scalable, automated judges. We prompt a diverse suite of ten LMMs to score the MMQAs from our benchmark. Each LMM judge is provided with a structured prompt containing the detailed multi-dimensional rubric (identical to the one used by human experts), and a required JSON output format. We then measure the alignment between the LMM-generated scores and the human ground truth. The complete prompt templates are provided in Appendix E.

## 5 PRELIMINARY SOLUTION

Building upon the validated reliability of automated judges, we present the complete Q-Mirror Agent. It is an autonomous system that orchestrates generation and refinement within a closed-loop workflow, as detailed in Algorithm 1. The details are as follows:

*1) The Planner Stage.* The Q-Mirror begins with the Planner, a module that produces $K$ candidate MMQAs from a given TQA. For each candidate, the Planner simultaneously 1) reformulates the textual component through a process that involves identifying parts best converted to visuals, replacing them with explicit placeholders, and rephrasing the context, and 2) generates a detailed visual description. This description is then immediately converted into a rendered image via a coupled T2I model, yielding a complete MMQA. Importantly, the Planner is feedback-aware: in subsequent iterations, it conditions its generation not only on the source TQA but also on structured revision signals provided by the Controller, enabling progressive improvement.

*2) The Evaluator Stage.* The resulting candidates are then subjected to the Evaluator, carried out by $M$ top-performing LMMs ranked in our benchmark (§4.3). The ensemble not only assigns rubric-based quality scores but also provides qualitative judgments on critical flaws such as semantic omissions, factual inaccuracies, or weak cross-modal alignment, together with prescriptive suggestions on how these issues can be refined in the next iteration. If the best candidate already exceeds the predefined quality threshold $\tau$, it is accepted as the final output. Otherwise, the collective feedback of the evaluators serves as the basis for refinement.

Table 3: Performance of model families (*the description generator and image generator from the same family, details in §6.1*)) on MMQA generation benchmark, top performance highlighted. The scores are all rescaled to a 0-100 range for presentation.

| Model Family | IL | IA | IC | SI | SQ | RE | CM | NE | TQ | AQ | SC | QT | AVG | RK |
|---|---|---|---|---|---|---|---|---|---|---|---|---|---|---|
| *Performance on Q-Mirror-Expert* | | | | | | | | | | | | | | |
| Doubao Family | 61.54 | 85.39 | 73.46 | 100.00 | 90.65 | 18.55 | 69.73 | 99.68 | 77.69 | 69.23 | 29.23 | 68.96 | 70.54 | 4 |
| Qwen2 Family | 76.15 | 73.08 | 74.62 | 99.03 | 83.23 | 51.92 | 78.06 | 97.74 | 72.46 | 71.46 | 33.08 | 68.69 | 73.28 | 2 |
| Grok Family | 62.31 | 75.48 | 68.90 | 99.03 | 85.48 | 60.40 | 81.64 | 90.97 | 75.00 | 69.23 | 26.92 | 65.53 | 71.37 | 3 |
| GPT Family | 71.54 | 80.77 | 76.15 | 100.00 | 88.71 | 71.94 | 86.88 | 98.07 | 80.00 | 69.23 | 26.92 | 68.55 | 76.33 | 1 |
| Qwen-VL Family | 64.96 | 71.47 | 68.22 | 100.00 | 80.72 | 51.95 | 77.56 | 98.70 | 75.35 | 59.69 | 27.09 | 65.21 | 69.81 | 5 |
| Q-Mirror Agent | 79.29 | 84.36 | 81.83 | 100 | 91.58 | 75.49 | 89.03 | 100 | 82.63 | 78.69 | 52.65 | 78.49 | 82.65 | / |
| *Performance on Q-Mirror-Grad* | | | | | | | | | | | | | | |
| Doubao Family | 88.39 | 87.42 | 87.90 | 100.00 | 90.77 | 45.19 | 78.65 | 100.00 | 79.36 | 79.03 | 30.00 | 68.96 | 77.55 | 2 |
| Qwen2 Family | 85.48 | 73.87 | 79.68 | 99.23 | 90.77 | 55.24 | 81.75 | 100.00 | 78.71 | 78.39 | 39.03 | 68.69 | 75.90 | 3 |
| Grok Family | 80.65 | 80.00 | 80.32 | 100.00 | 93.08 | 60.85 | 84.64 | 97.69 | 81.61 | 75.16 | 32.26 | 65.53 | 75.70 | 4 |
| GPT Family | 83.23 | 96.13 | 89.68 | 100.00 | 89.23 | 72.69 | 87.31 | 100.00 | 77.74 | 80.32 | 29.44 | 68.55 | 80.52 | 1 |
| Qwen-VL Family | 85.44 | 72.25 | 78.84 | 97.67 | 91.47 | 60.41 | 83.19 | 99.23 | 81.43 | 72.96 | 28.66 | 65.21 | 74.69 | 5 |
| Q-Mirror Agent | 87.98 | 94.63 | 91.30 | 100 | 92.15 | 82.65 | 91.60 | 100 | 84.97 | 86.35 | 57.67 | 78.49 | 86.27 | / |
| *Performance on Q-Mirror-(Expert+Grad)* | | | | | | | | | | | | | | |
| Doubao Family | 69.47 | 85.99 | 77.73 | 100.00 | 90.68 | 26.42 | 72.37 | 99.77 | 78.18 | 72.13 | 29.46 | 72.10 | 73.87 | 4 |
| Qwen2 Family | 78.91 | 73.31 | 76.11 | 99.09 | 85.45 | 52.90 | 79.15 | 98.41 | 74.31 | 73.51 | 34.84 | 74.03 | 76.19 | 2 |
| Grok Family | 67.73 | 76.82 | 72.27 | 99.32 | 87.73 | 60.53 | 82.53 | 92.95 | 76.95 | 70.98 | 28.50 | 71.68 | 75.11 | 3 |
| GPT Family | 74.99 | 85.31 | 80.15 | 100.00 | 88.86 | 72.16 | 87.01 | 98.64 | 79.33 | 72.51 | 27.67 | 71.88 | 78.90 | 1 |
| Qwen-VL Family | 71.01 | 71.70 | 71.36 | 99.31 | 83.89 | 54.45 | 79.22 | 98.85 | 77.15 | 63.61 | 27.55 | 70.57 | 73.40 | 5 |
| Q-Mirror Agent | 81.86 | 87.40 | 84.63 | 100 | 91.75 | 77.61 | 89.79 | 100 | 83.32 | 80.95 | 54.13 | 82.25 | 85.22 | / |

*3) The Controller Stage.* Finally, the Controller synthesizes the feedback of the ensemble into actionable, structured revision instructions. Rather than aggregating scores mechanically, the Controller prioritizes high-severity issues (e.g., factual errors) over superficial aspects (e.g., stylistic variation), thereby guiding the Planner toward substantive improvements. The refined instructions are then returned to the Planner, which regenerates a new batch of candidates. This cycle of *generation–evaluation–revision* continues until a candidate surpasses the quality threshold or the maximum number of iterations $N$ is reached.

## 6 EXPERIMENTS

A series of experiments are conducted to validate the proposed framework, designed to answer three key questions: 1) How do the generation models perform on the TQA-to-MMQA transformation task? 2) How reliable are leading understanding models as MMQA quality judges compared to human experts? 3) How effectively does the proposed Q-Mirror agent improve MMQA quality?

### 6.1 EXPERIMENTAL SETUP

**Models Families for MMQA Generation.** For MMQA generation, we benchmark five diverse model families, including proprietary and prominent open-source leaders, including the **GPT Family** (*GPT-4.1-2025-04-14* (OpenAI, 2025a), *GPT-Image-1* (OpenAI, 2025b)), the **Doubao Family** (*Doubao-1-5-Thinking-Pro-250415* (ByteDance, 2024), *Doubao-Seedream-3-0-T2I-25041* (Gao et al., 2025)), the **Qwen2 Family** (*Qwen2-VL-72B-Instruct* (Wang et al., 2024b), *Wanx2.1-T2I-Plus* (Wan et al., 2025)), the **Grok Family** (*Grok-4* (xAI, 2025b), *Grok-2-Image* (xAI, 2025a)), and **Qwen-VL Family** (*Qwen-VL-Max-Latest* (Bai et al., 2023), *Qwen-Image* (Wu et al., 2025)). By adopting intra-family model pairing to reduce architectural inconsistencies and modal gaps, this strategy enables a more accurate comparison of different model families in scientific T2I generation, particularly in evaluating semantic alignment and content faithfulness.

**Models for MMQA Evaluation.** For the MMQA quality judge evaluation, we assess ten powerful LMMs including *Qwen2-VL-72B-Instruct* (Wang et al., 2024b), *Claude-Sonnet-4* (Anthropic, 2025), *DeepSeek-VL2* (Wu et al., 2024), *Doubao-1-5-Vision-Pro-250328* (ByteDance, 2024), *Gemini-2.5-Pro* (Team et al., 2023), *GPT-4.1-2025-04-14* (OpenAI, 2025a), *Grok-4* (xAI, 2025b), *Llama-3.2-90B-Vision* (Grattafiori et al., 2024), *Pixtral-12B* (Agrawal et al., 2024), and *O3* (OpenAI, 2025c), to cover a range of architectures. All models are evaluated against our multi-dimensional rubric (§5).

**Q-Agent Parameters Setup.** In practice, we set $\tau = 80.0$ (the acceptance threshold score), $M = 3$ (the number of top-performing LMMs in ensemble), $N = 5$ (the maximum number of refinement iterations), and $K = 4$ (the number of candidates generated per iteration) for Q-Mirror, balancing efficiency and reliability. In addition to reporting the AVG, the *pass rate* is also defined as the proportion of MMQAs whose AVG exceeds the acceptance threshold $\tau = 80.0$. This metric reflects not only average improvements but also the robustness of the system in consistently producing high-quality outputs. Moreover, the Q-Mirror agent *Planner* is instantiated with *GPT-Image-1*). The

Table 4: Performance on MMQA evaluation benchmark. 10 LMMs are evaluated by measuring the alignment of their scores with human expert annotations, with the best performance highlighted. The scores are all rescaled to a 0-100 range for presentation.

| Model | IL | IA | IC | SI | SQ | RE | CM | NE | TQ | AQ | SC | QT | AVG | RK |
|---|---|---|---|---|---|---|---|---|---|---|---|---|---|---|
| *Judge Alignment on Q-Mirror-Expert* | | | | | | | | | | | | | | |
| Qwen2-VL-72B-Instruct | 62.31 | 31.94 | 47.12 | 95.81 | 47.42 | 26.13 | 56.45 | 39.03 | 82.58 | 31.29 | 48.39 | 50.32 | 51.20 | 8 |
| Claude-Sonnet-4 | 71.93 | 46.92 | 59.42 | 96.77 | 30.32 | 38.06 | 55.05 | 53.42 | 79.19 | 47.90 | 46.45 | 56.74 | 57.04 | 5 |
| DeepSeek-VL2 | 60.77 | 31.54 | 46.15 | 91.85 | 20.32 | 24.19 | 45.45 | 36.45 | 89.81 | 37.42 | 55.81 | 43.23 | 44.77 | 10 |
| Doubao-1.5-Vision-Pro | 65.38 | 37.15 | 51.27 | 97.92 | 61.61 | 26.77 | 62.10 | 32.26 | 59.29 | 39.35 | 57.10 | 47.00 | 52.81 | 6 |
| Gemini-2.5-Pro | 72.69 | 42.46 | 57.58 | 99.68 | 50.97 | 30.32 | 60.32 | 52.06 | 87.61 | 38.28 | 51.61 | 57.39 | 58.33 | 4 |
| GPT-4.1 | 70.00 | 39.35 | 54.68 | 96.77 | 45.48 | 34.19 | 58.82 | 46.45 | 96.45 | 57.10 | 49.32 | 62.33 | 58.98 | 2 |
| Grok-4 | 71.54 | 47.74 | 59.64 | 95.48 | 26.77 | 42.90 | 55.05 | 54.62 | 83.87 | 55.48 | 68.71 | 65.67 | 60.68 | 1 |
| Llama-3.2-90B-Vision | 63.08 | 41.55 | 52.31 | 89.92 | 29.35 | 27.74 | 49.01 | 48.06 | 87.71 | 40.32 | 38.39 | 53.62 | 51.84 | 7 |
| Pixtral-12B | 60.50 | 39.26 | 49.88 | 90.58 | 34.52 | 20.97 | 48.69 | 44.85 | 83.87 | 26.77 | 40.65 | 49.03 | 49.18 | 9 |
| O3 | 72.31 | 52.03 | 62.17 | 98.54 | 42.10 | 24.74 | 55.13 | 47.85 | 82.90 | 45.16 | 57.74 | 58.41 | 58.55 | 3 |
| Q-Mirror Agent | 79.50 | 69.95 | 74.73 | 99.83 | 69.75 | 54.03 | 74.54 | 67.05 | 98.61 | 60.91 | 66.42 | 73.25 | 74.08 | / |
| *Judge Alignment on Q-Mirror-Grad* | | | | | | | | | | | | | | |
| Qwen2-VL-72B-Instruct | 74.10 | 43.85 | 58.97 | 100.00 | 70.77 | 33.08 | 67.95 | 42.77 | 88.46 | 35.71 | 49.31 | 54.06 | 59.70 | 7 |
| Claude-Sonnet-4 | 84.68 | 51.94 | 68.31 | 100.00 | 73.08 | 40.77 | 71.28 | 51.77 | 80.77 | 49.23 | 49.54 | 57.83 | 65.01 | 3 |
| DeepSeek-VL2 | 77.39 | 36.13 | 56.76 | 92.52 | 67.25 | 36.15 | 65.31 | 46.29 | 89.38 | 41.54 | 56.54 | 48.12 | 55.87 | 10 |
| Doubao-1.5-Vision-Pro | 76.55 | 42.77 | 59.66 | 100.00 | 70.77 | 26.15 | 65.64 | 45.38 | 63.08 | 44.65 | 56.92 | 55.13 | 59.64 | 8 |
| Gemini-2.5-Pro | 83.55 | 47.85 | 65.70 | 99.68 | 63.85 | 26.92 | 63.48 | 52.23 | 88.54 | 37.69 | 56.92 | 58.85 | 62.29 | 5 |
| GPT-4.1 | 84.84 | 63.85 | 74.34 | 100.00 | 71.54 | 37.69 | 69.74 | 57.54 | 97.69 | 62.31 | 60.15 | 69.42 | 71.00 | 1 |
| Grok-4 | 82.71 | 52.92 | 67.82 | 96.15 | 65.38 | 37.77 | 61.58 | 57.26 | 91.77 | 56.15 | 70.77 | 68.99 | 63.41 | 4 |
| Llama-3.2-90B-Vision | 73.52 | 42.31 | 57.91 | 91.87 | 56.11 | 46.92 | 64.97 | 49.23 | 88.69 | 56.92 | 40.69 | 58.88 | 60.42 | 6 |
| Pixtral-12B | 74.52 | 46.45 | 60.48 | 91.17 | 68.16 | 26.92 | 62.08 | 46.68 | 86.89 | 27.65 | 51.62 | 53.21 | 58.05 | 9 |
| O3 | 86.87 | 55.38 | 71.13 | 98.46 | 68.77 | 37.10 | 68.11 | 60.97 | 86.15 | 52.73 | 59.23 | 64.77 | 67.68 | 2 |
| Q-Mirror Agent | 81.86 | 87.40 | 84.63 | 100 | 91.75 | 77.61 | 89.79 | 100 | 83.32 | 80.95 | 54.13 | 82.25 | 85.22 | / |
| *Judge Alignment on Q-Mirror-(Expert+Grad)* | | | | | | | | | | | | | | |
| Qwen2-VL-72B-Instruct | 65.79 | 35.45 | 50.62 | 97.05 | 54.32 | 28.18 | 59.85 | 40.14 | 84.32 | 32.60 | 48.66 | 51.43 | 53.71 | 8 |
| Claude-Sonnet-4 | 75.69 | 48.40 | 62.05 | 97.73 | 42.95 | 38.86 | 59.85 | 52.93 | 79.66 | 48.30 | 47.36 | 57.06 | 59.39 | 5 |
| DeepSeek-VL2 | 65.68 | 32.89 | 49.29 | 92.04 | 34.19 | 27.73 | 51.32 | 39.36 | 89.68 | 38.64 | 56.02 | 55.92 | 52.55 | 9 |
| Doubao-1.5-Vision-Pro | 68.68 | 38.81 | 53.75 | 98.54 | 64.32 | 26.59 | 63.15 | 36.14 | 60.41 | 40.92 | 57.05 | 48.63 | 54.52 | 6 |
| Gemini-2.5-Pro | 75.90 | 44.05 | 59.98 | 99.68 | 54.77 | 29.32 | 61.26 | 49.73 | 87.89 | 38.11 | 53.18 | 57.82 | 59.50 | 4 |
| GPT-4.1 | 74.38 | 46.59 | 60.49 | 97.73 | 53.18 | 35.23 | 62.05 | 49.73 | 96.82 | 58.64 | 52.52 | 64.43 | 62.53 | 2 |
| Grok-4 | 74.84 | 49.27 | 62.06 | 95.68 | 38.18 | 41.39 | 58.42 | 55.40 | 86.20 | 55.68 | 69.32 | 66.65 | 62.80 | 1 |
| Llama-3.2-90B-Vision | 66.16 | 41.77 | 53.97 | 90.50 | 37.26 | 33.41 | 53.72 | 44.82 | 88.00 | 45.23 | 39.07 | 55.18 | 54.38 | 7 |
| Pixtral-12B | 64.64 | 41.38 | 53.01 | 90.76 | 44.45 | 22.73 | 52.65 | 45.39 | 84.76 | 27.03 | 43.89 | 50.27 | 51.80 | 10 |
| O3 | 76.61 | 53.02 | 64.82 | 98.52 | 49.98 | 28.39 | 58.96 | 51.72 | 83.86 | 47.40 | 58.18 | 60.29 | 61.25 | 3 |
| Q-Mirror Agent | 81.87 | 69.47 | 75.67 | 99.88 | 71.98 | 52.03 | 74.63 | 65.49 | 98.55 | 59.65 | 68.08 | 72.94 | 74.27 | / |

Figure 3: Performance comparison for the top-3 judge models and Q-Mirror *Evaluator* (judge ensemble group). In each chart, the **red line** shows the ratio between the overall average of all ten judges and the average of the top three (Avg10/AvgTop-3). The **blue line** indicates the relative performance of the current judge (or ensemble) compared with the top-three average (Scorecurrent/AvgTop-3).

Evaluator module is an ensemble of the top-three performing judges identified in the evaluation benchmark: *Grok-4*, *GPT-4.1-2025-04-14*, and *O3*. The specific prompts used for generation and evaluation are detailed in Appendix C, Appendix D, and Appendix E.

## 6.2 FINDINGS

**Finding for MMQA Generation.** As shown in Table 3, the **GPT Family** achieves the highest overall quality with an average score of 78.90, which highlights its strong capability for scientific T2I tasks and underscores its potential as the base module for Q-Mirror. Secondly, a clear trend emerges across all model families: performance on Q-Mirror-Grad consistently surpasses that on Q-Mirror-Expert. This gap might indicate that current models handle structured knowledge with standard visual patterns more effectively than expert-level concepts that lack canonical visualizations. Thirdly, a consistent pattern is observed across all model families: they achieve near-perfect scores on the SI dimension (i.e., whether the question can be solved from the image alone). This indicates that the generated scientific images are reasonably self-contained without revealing the answer

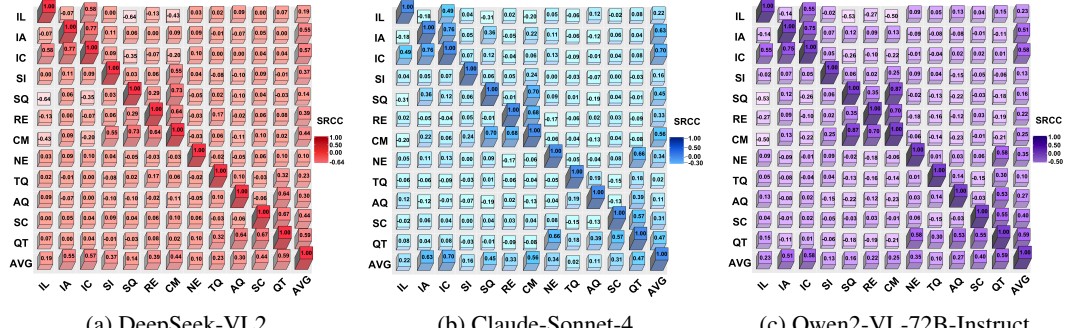

|                  |                     |                           |
| ---------------- | ------------------- | ------------------------- |
| (a) DeepSeek-VL2 | (b) Claude-Sonnet-4 | (c) Qwen2-VL-72B-Instruct |

Figure 4: Dimensional SRCC correlations of MMQA evaluation, following the setting of Redundancy Principle (Zhang et al., 2025c). Higher and darker bars indicate stronger agreement between the corresponding dimensions.

directly. In contrast, their performance on the SC dimension (i.e., whether the image is plausible and scientifically sound) is consistently poor. This stark disparity highlights a significant limitation of current T2I models, particularly in rendering legible text and ensuring scientific plausibility, indicating a substantial gap that must be bridged for practical application.

**Findings for MMQA Evaluation.** Table 4 reveals two key observations. First, judge performance is clearly stratified: a top tier led by *Grok-4* achieves the highest agreement with human annotations (62.80%), whereas several weaker models fall below 55%. Second, task difficulty varies with the abstraction level. All judges align more closely with humans on Q-Mirror-Grad than on Q-Mirror-Expert. For instance, *Claude-Sonnet-4* improves from 58.98% on Expert to 71.00% on Grad, reflecting that graduate-level questions admit more objective criteria, while expert-level questions involve subjective reasoning, which increase divergence. Third, as illustrated in Figure 4, the correlations among the major dimensions (e.g., IC, CM, QT) are low, which validates the overall non-redundancy of the benchmark design (Zhang et al., 2025c). Furthermore, this result indirectly affirms the effectiveness of the proposed quality principles.

### 6.3 PERFORMANCE OF Q-MIRROR

The findings above highlight the risk of relying on a single judge, as its biases would directly affect evaluation, especially for complex scientific content. To mitigate this, we adopt a Judge Ensemble for Q-Mirror composed of the top three models *Grok-4*, *GPT-4.1-2025-04-14*, and *O3* (*judge performance shown in Figure 3*), which aggregates diverse perspectives, reduces model-specific bias. In total, the judge ensemble provides more stable and reliable feedback across dimensions, mitigates single-model biases, and guarantees closer alignment with human judgment.

In addition, Table 3 demonstrates that the Q-Mirror agent consistently outperforms all baseline families, improving the overall average score from 78.90 to 85.22 and raising the pass rate from 72% to 95% (*definition in §6.1*). This demonstrates that Q-Mirror not only enhances overall quality but also ensures greater stability and consistency in producing high-quality MMQAs. Further analysis reveals critical insights: While models excel at solvability from images alone, they consistently struggle with scientific plausibility, underscoring the difficulty of rendering domain-accurate visuals. Moreover, the greater improvements on expert-level tasks suggest that Q-Mirror is particularly valuable for challenging, less standardized scientific problems.

## 7 CONCLUSION

This work addresses the critical scarcity of multi-modal scientific QA pairs by introducing Q-Mirror, a systematic framework for converting TQAs into high-quality MMQAs. Our contributions are threefold. First, we establish a **transformation pipeline and comprehensive quality rubric** that define principles for effective SMQ-to-MMQA transformation. Second, we introduce **two extensive benchmarks** for MMQA generation and evaluation, revealing distinct strengths and weaknesses in current state-of-the-art models. Third, we implement the **Q-Mirror**, an autonomous system that integrates generation and judgment through iterative refinement. The agent achieves an average quality score of 85.22 and a 95% pass rate, demonstrating the efficacy of our feedback-driven architecture. By unlocking the multi-modal potential of text-based corpora, Q-Mirror provides a scalable and cost-effective approach to advance scientific AI evaluation and support the development of next-generation reasoning models.

## 8 ETHICS STATEMENT

This work adheres to the ICLR Code of Ethics. In this study, no human subjects or animal experimentation was involved. All datasets used, including Q-Mirror-(Expert+Grad), are sourced in compliance with relevant usage guidelines, ensuring no violation of privacy. We have taken care to avoid any biases or discriminatory outcomes in our research process. No personally identifiable information is used, and no experiments are conducted that could raise privacy or security concerns. We are committed to maintaining transparency and integrity throughout the research process.

## 9 REPRODUCIBILITY STATEMENT

We have made every effort to ensure that the results presented in this paper are reproducible. We guarantee that all relevant code and datasets will be made publicly available, thereby enabling the research community to replicate and verify our findings. The experimental setup, including training steps, model configurations, and hardware details, is described in detail in the paper. We have also provided a full description of Q-Mirror, to assist others in reproducing our experiments.

Additionally, datasets used in the paper are publicly available, ensuring consistent and reproducible evaluation results.

We believe these measures will enable other researchers to reproduce our work and further advance the field.

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
