## A LLM USAGE

Large Language Models (LLMs) are used to aid in the writing and polishing of the manuscript. Specifically, we use an LLM to assist in refining the language, improving readability, and ensuring clarity in various sections of the paper. The model helps with tasks such as sentence rephrasing, grammar checking, and enhancing the overall flow of the text.

It is important to note that the LLM is not involved in the ideation, research methodology, or experimental design. All research concepts, ideas, and analyses are developed and conducted by the authors. The contributions of the LLM are solely focused on improving the linguistic quality of the paper, with no involvement in the scientific content or data analysis.

The authors take full responsibility for the content of the manuscript, including any text generated or polished by the LLM. We ensure that the LLM-generated text adheres to ethical guidelines and does not contribute to plagiarism or scientific misconduct.

## B DATASET DETAILS

This section presents the methodology for constructing the Q-Mirror-Expert and Q-Mirror-Grad datasets. The design emphasizes quality, diversity, and evaluative power to enable rigorous multimodal assessment.

**Phase 1: Candidate Pool Construction** We begin by sampling candidate TQAs from multiple authoritative scientific benchmarks(Rein et al., 2024; Wang et al., 2025c) to ensure disciplinary breadth and diversity. An LMM automatically filters questions, discarding those without clear visualizable components. Human experts then review the remaining items to verify their scientific accuracy, refine their formulations, and guarantee balanced coverage across 22 scientific disciplines. This process yields a curated pool of 440 high-quality TQAs.

**Phase 2: Difficulty-based Partitioning** The curated pool is then partitioned into two subsets according to difficulty, determined jointly by dataset-provided difficulty labels and expert judgment.

Q-MIRROR-EXPERT (EXPERT-LEVEL SUBSET) This subset consists of 310 questions (approximately 70% of the pool). It emphasizes **novelty, abstraction, and creativity**, including frontier and interdisciplinary concepts, highly abstract problems, and cases where no canonical visual representation exists. It thus evaluates higher-order reasoning and creative inference.

Q-MIRROR-GRAD (GRADUATE-LEVEL SUBSET) This subset contains 130 questions (approximately 30% of the pool). It emphasizes **structure, established knowledge, and accuracy**, covering mature scientific concepts with canonical visual forms such as molecular structures, circuit diagrams, or biological processes. It thereby measures fidelity in reproducing structured scientific information. The GPQA IDs included in this subset are listed in Table 5.

Table 5: GPQA Question IDs Included in Q-Mirror-Grad.

| GPQA IDs | | | | | | | | | | | | | | |
|---|---|---|---|---|---|---|---|---|---|---|---|---|---|---|
| 0 | 1 | 2 | 4 | 7 | 8 | 9 | 12 | 13 | 14 | 15 | 16 | 17 | 18 | 19 |
| 20 | 22 | 23 | 24 | 25 | 27 | 28 | 30 | 32 | 33 | 35 | 36 | 37 | 38 | 40 |
| 41 | 42 | 43 | 44 | 45 | 46 | 47 | 48 | 49 | 50 | 52 | 54 | 55 | 56 | 58 |
| 60 | 61 | 62 | 63 | 64 | 66 | 67 | 68 | 71 | 72 | 74 | 75 | 76 | 77 | 78 |
| 80 | 81 | 82 | 84 | 85 | 86 | 87 | 88 | 89 | 90 | 91 | 92 | 93 | 94 | 95 |
| 96 | 97 | 98 | 99 | 100 | 101 | 102 | 105 | 107 | 109 | 110 | 112 | 113 | 114 | 115 |
| 116 | 117 | 118 | 120 | 121 | 123 | 125 | 126 | 127 | 128 | 130 | 131 | 132 | 138 | 140 |
| 141 | 142 | 144 | 145 | 147 | 148 | 149 | 150 | 152 | 153 | 154 | 155 | 157 | 158 | 159 |
| 160 | 161 | 162 | 163 | 168 | 169 | 174 | 175 | 191 | 194 | | | | | |

## C HUMAN EVALUATION PROTOCOL

This section details the protocol employed for human expert evaluation, which serves as the gold standard for assessing MMQA quality and benchmarking LMMs as judges.

Figure 5: Custom-built web interface for human evaluation. Annotators view the original TQA and the transformed MMQA (text and image), then score the MMQA according to fine-grained metrics. The interface enforces randomized assignment, and automatically exports results in JSON format to ensure independence and reproducibility.

**Annotator Background and Training** We recruit a panel of 50 annotators, all of whom are Ph.D. students in STEM disciplines with undergraduate training in fields such as physics, chemistry, biology, or mathematics. Each annotator undergoes a standardized 4-hour training session, which includes:

- A comprehensive review of the formal MMQA quality rubric and its underlying principles (Section 3.2).

- An analysis of high- and low-quality MMQA examples illustrating common pitfalls such as information loss, redundant visuals, and factual inaccuracies.

- Practice rounds on 20 sample MMQAs, followed by a group discussion and calibration session to ensure consistent interpretation and application of scoring guidelines.

Periodic calibration sessions are further conducted to maintain consistency throughout the evaluation process.

**Evaluation Procedure** Each model-generated MMQA is randomly assigned to two annotators, with a third annotator included in cases of disagreement. Annotators use a custom-built web interface (Figure 5) that presents the original TQA alongside the transformed MMQA (text and image). The interface enforces randomized assignment and prevents annotators from viewing others' scores, thereby ensuring independence. Annotators score each MMQA according to 11 fine-grained metrics grouped under three principles: Information Consistency (IC), Cross-Modal Integration (CM), and Standalone Quality (QT). For any score below the maximum, annotators also provide brief textual justifications.

PRINCIPLE 1: INFORMATION CONSISTENCY (IC)

This principle measures the semantic fidelity of the transformed MMQA relative to the source TQA.

*Metric 1.1: Information Loss (IL / $p_-$)*

- Definition: Assesses whether any critical information from the original TQA is missing in the MMQA.

- Scoring (Binary): 1 = No information loss, 0 = Critical information missing.

*Metric 1.2: Information Addition (IA / $p_+$)*

- Definition: Assesses whether spurious or non-inferable information is introduced.

- Scoring (Binary): 1 = No spurious information, 0 = Spurious information added.

PRINCIPLE 2: CROSS-MODAL INTEGRATION (CM)

This principle evaluates whether text and image provide complementary and indispensable information.

*Metric 2.1: Solvability with Image (SI / $p_s(I)$)*

- Definition: Tests whether the problem can be solved using the image alone.

- Scoring (Binary): 1 = Not solvable with image alone (desirable), 0 = Solvable with image alone.

*Metric 2.2: Solvability with Question (SQ / $p_s(T)$)*

- Definition: Tests whether the problem can be solved using the text alone.

- Scoring (Binary): 1 = Not solvable with text alone (desirable), 0 = Solvable with text alone.

*Metric 2.3: Redundancy-Synergy (RE / $f_s$)*

- Definition: Measures the degree of information overlap between text and image.

- Scoring (Ordinal): 1 = Synergistic, no redundancy, 50 = Partial redundancy, 0 = Complete redundancy.

PRINCIPLE 3: STANDALONE QUALITY (QT)

This principle assesses the intrinsic quality of each modality independently.

*Metric 3.1: Natural Expression (NE / $p_{nat}$)*

- Definition: Evaluates the fluency, coherence, and grammatical quality of the text.

- Scoring (Binary): 1 = Fluent and correct, 0 = Contains significant linguistic issues.

*Metric 3.2: Technical Quality (TQ / $p_{tech}$)*

- Definition: Evaluates whether the image is technically sound and artifact-free.

- Scoring (Binary): 1 = Technically flawless, 0 = Contains noticeable defects.

*Metric 3.3: Aesthetic Quality (AQ / $p_{aes}$)*

- Definition: Evaluates the composition, layout, and visual appeal of the image.

- Scoring (Binary): 1 = Well-composed and professional, 0 = Poorly composed or unappealing.

*Metric 3.4: Semantic Clarity (SC / $p_{sem}$)*

- Definition: Evaluates whether the image is unambiguous, scientifically accurate, and factually correct.

- Scoring (Binary): 1 = Accurate and unambiguous, 0 = Contains errors or ambiguity.

**Inter-Annotator Agreement (IAA)**   To quantify the reliability of human judgments, we compute Krippendorff's Alpha ($\alpha$) across all metrics prior to consensus resolution. We achieve a high average agreement of $\alpha = 0.82$, validating the robustness of the protocol and the reliability of human annotations as a gold standard. The evaluation interface (Figure 5) further supports this process by enforcing independent scoring and standardized output, thereby minimizing bias and ensuring reproducibility.

## D   PROMPTS FOR GENERATION

This section provides the full set of prompt templates used for instructing the MMQA generation models. This template is used to guide the multi-modal question modification assistant to generate image-related fields.

*System Prompt:*
*#System: You are a professional multi-modal question modification assistant. Your task is to generate the image-related fields for the given multiple-choice question without modifying the original question, options, or answer.*
*The modified question must meet the following criteria:*
*- Both text and image are required to answer the question correctly. Neither the image alone nor the modified text alone can fully solve the question.*
*- The modified text must be clear, high-quality, and complete, but it must be understood in conjunction with the image.*
*- The image description must be specific, clear, and comprehensive (while strictly less than 1024 characters) to ensure high-quality image generation and clarity of information.*
*- The overall information content and difficulty of the question must remain consistent before and after modification.*
*- The visual information should be removed from the modified text, and instead, it should be represented in the image.*
*- For example, if the question previously described an energy difference between two states, the visual representation of the energy difference should be shown in the image, while the text should simply provide the context or question that requires the image for full understanding.*
*- The image description should be as detailed as possible (while remaining under 1024 characters), providing specific instructions for creating the image. The description should cover all the visual*

*aspects required to convey the context of the question, ensuring a high-quality visual representation.*
*- The image style and composition must be neutral and objective. The visual style must be appropriate for the question's subject area (e.g., a clean diagram for science, an antique-style illustration for history). Critically, the image's composition, color, and layout must not use artistic techniques (such as special highlights, pointing arrows, or making elements related to one option more prominent) to unintentionally hint at or guide towards the correct answer, to ensure the question's fairness.*
*- The text and visual information between the modified_question and image_description should not overlap. The image should visually represent what the text cannot explain or clarify, avoiding any duplication of the same information in both the text and the image.*

Please return the following JSON format: {
"images": [
{
"placeholder": "[IMAGE_1]",
"original_text": "The original text that should be replaced",
"image_description": "A detailed description (must be less than 1024 characters) of the image to be generated" }
],
"modified_question": "The modified question text, containing a placeholder [IMAGE_1] for the image position. Ensure that the visual content is removed from the text and only context remains.",
"explanation": "A brief explanation of why both the image and text are needed."
}

## E    PROMPTS FOR LMM QUALITY JUDGE

This section provides the full set of prompt templates used for instructing the LMM quality judges. The prompt to instruct an LMM to act as a quality judge for an MMQA is designed to elicit structured scores and justifications based on our rubric. This template is used to guide the professional exam quality reviewer to evaluate multi-modal questions across different stages.

**Stage 1: Evaluate information loss**

*#System: You are a professional exam quality reviewer. Your task is to evaluate whether there is any information loss between the original and transformed question. Respond strictly as instructed.*
*Evaluation schema:  a) Is there any critical information loss before and after the modification?*
*Original question: "item['question']"*
*Modified question: "item['modified_question']"*
*Added image: "item['image_url']"*

Please answer strictly in the following format:
{ a) [yes/no]
l) [your comments explaining the reasoning behind your answer to a)] }

**Stage 2: Evaluate information addition**

*#System: You are a professional exam quality reviewer. Your task is to evaluate whether there is any information addition between the original and transformed question. Respond strictly as instructed.*
*Evaluation schema:  b) Is there any critical information gain before and after the modification?*
*Modified question: "item['modified_question']"*
*Added image: "item['image_url']"*

Please answer strictly in the following format:
{b) [yes/no]
l) [your comments explaining the reasoning behind your answer to b)] }

**Stage 3: Evaluate solvability with image only**

*#System: You are a professional exam quality reviewer.  Your task is to evaluate whether the transformed question can be solved using only the image and options. Respond strictly as instructed.*
*Evaluation schema:  c) For the transformed question, looking only at the image and options, can you solve the problem?*

*Added image: "item['image_url']"*
*(if have) Options: "item['question']"*

Please answer strictly in the following format:
{c) [yes/no]
l) [your comments explaining the reasoning behind your answer to c)] }

**Stage 4: Evaluate solvability with text only**

*#System: You are a professional exam quality reviewer. Your task is to evaluate whether the transformed question can be solved using only the modified question text and options. Respond strictly as instructed.*
*Evaluation schema: d) For the transformed question, looking only at the transformed question and options, can you solve the problem?*
*Modified question: "item['modified_question']"*
*(if have) Options: "item['question']"*

Please answer strictly in the following format:
{d) [yes/no]
l) [your comments explaining the reasoning behind your answer to d)] }

**Stage 5: Evaluate information redundancy**

*#System: You are a professional exam quality reviewer. Your task is to evaluate the level of redundant information between the image and text in the transformed question. Respond strictly as instructed.*
*Evaluation schema:  e) For the transformed question, is there any redundant information between the image and text (0: no redundancy, 1: partial, 2: complete)?*
*Original question: "item['question']"*
*Modified question: "item['modified_question']"*
*Added image: "item['image_url']"*

Please answer strictly in the following format:
{e) [0/1/2] (0: no redundancy, 1: partial redundancy, 2: complete redundancy)
l) [your comments explaining the reasoning behind your answer to e)] }

**Stage 6: Evaluate logical consistency and language naturalness**

*#System: You are a professional exam quality reviewer. Your task is to evaluate whether the transformed question is logically consistent and has natural language. Respond strictly as instructed.*
*Evaluation schema:  f) Is the transformed question logically consistent and the language natural?*
*Modified question: "item['modified_question']"*

Please answer strictly in the following format:
{ f) [yes/no]
l) [your comments explaining the reasoning behind your answer to f)] }

**Stage 7: Evaluate image technical quality**

*#System: You are a professional exam quality reviewer. Your task is to evaluate whether the technical quality of the image meets the requirements. Respond strictly as instructed.*
*Evaluation schema: h1) For the transformed question, does the technical quality of the image (noise, fractures, naturalness, distortion) meet the requirements?*
*Added image: "item['image_url']"*

Please answer strictly in the following format:
{ h1) [yes/no]
l) [your comments explaining the reasoning behind your answer to h1)] }

**Stage 8: Evaluate image aesthetic quality**

*#System: You are a professional exam quality reviewer. Your task is to evaluate whether the aesthetic quality of the image meets the requirements. Respond strictly as instructed.*
*Evaluation schema: h2) For the transformed question, does the aesthetic quality of the image (composition and design, visual elements and structure, technical execution, originality and creativity, theme and communication, emotion and audience response) meet the requirements?*
*Added image: "item['image_url']"*

Please answer strictly in the following format:
{ h2) [yes/no]
l) [your comments explaining the reasoning behind your answer to h2)] }

**Stage 9: Evaluate semantical clarity**

*#System: You are a professional exam quality reviewer. Your task is to evaluate whether the image is clear and conforms to objective laws. Respond strictly as instructed.*
*Evaluation schema: i) For the transformed question, is the image plausible (text is reasonable, conforms to objective laws)?*
*Added image: "item['image_url']"*

Please answer strictly in the following format:
{ i) [yes/no]
l) [your comments explaining the reasoning behind your answer to i)] }

## F  EXPERIMENTS

### F.1  EXPERIMENT ANALYSIS.

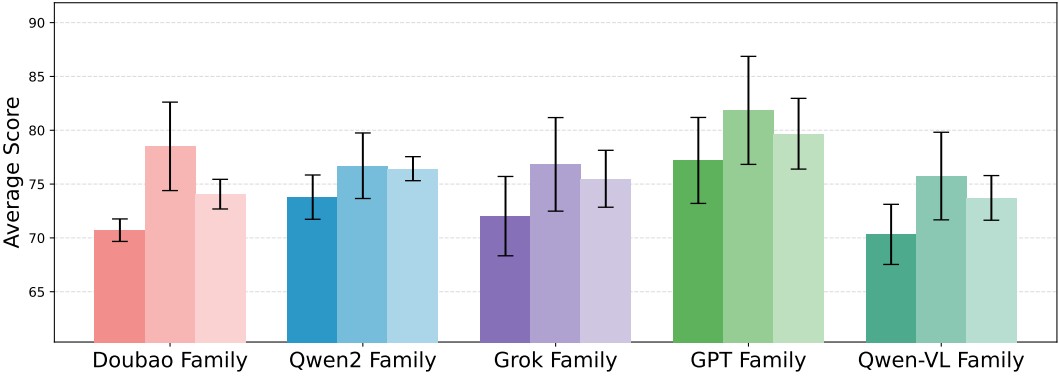

Figure 6: Robustness performance on MMQA generation Benchmark. Each bar shows the mean AVG score with variance, and the three bars from left to right indicate results on Q-Mirror-(Expert+Grad), Q-Mirror-Expert, and Q-Mirror-Grad.

**Findings for MMQA Generation.** Examining per-metric behavior (see §3.2.2), we observe consistently strong modal interdependence, reflected in high SI and SQ scores across model families. This confirms that our TQA-to-MMQA transformation enforces genuine cross-modal reasoning rather than allowing text-only shortcuts. By contrast, factual fidelity in the image space (SC) remains the principal limitation on the Expert subset, where mislabeled components and subtle schematic inaccuracies frequently occur. These findings indicate that one-shot generation is suboptimal for expert-level MMQA: models align modalities well but fail to ground fine-grained visual details reliably. This trend is further illustrated in Figure 6, which compares performance across model families under varying weight assignments for the three core evaluation dimensions (IC, CM, QT). The results show that while models remain stable under changes emphasizing modal interdependence (CM), their scores drop significantly when factual accuracy (IC/SC) receives higher weight, especially

*Original:* Fill in the blank: Given four elements A, B, C, and D, where A is a fourth-period element that can form a 1:1 atomic ratio compound with element D. B is a fourth-period d-block element with the highest positive oxidation state of +7. C is in the same period as B and has the same positive oxidation state. D is the element with the highest electronegativity among all elements. A. Fill in the table below | Element | Valence electron configuration | Period | Group | Metal or non-metal | |---------|-------------------------------|--------|-------|-----------------| | A | ( ) | ( ) | ( ) | ( ) | | B | ( ) | ( ) | ( ) | ( ) | | C | ( ) | ( ) | ( ) | ( ) | | D | ( ) | ( ) | ( ) | ( ) | B. Order the four elements by electronegativity from high to low ( ).

*Modified:* Fill in the blank: Given four elements A, B, C, and D, where A is a fourth-period element that can form a 1:1 atomic ratio compound with element D. B is a fourth-period d-block element with the highest positive oxidation state of +7. C is in the same period as B and has the same positive oxidation state. D is the element with the highest electronegativity among all elements. A. Fill in the table shown in [IMAGE_1]. B. Order the four elements by electronegativity from high to low ( ).

*Answer:*
A: $4s^1$, 4th, IA, Metal;
B: $3d^5 4s^2$, 4th, VIIB, Metal;
C: $3d^{10} 4s^1$, 4th, IB, Metal;
D: $2s^2 2p^5$, 2nd, VIIA, Non-metal;
D > A > C > B

*Analysis:*
1) The drawing location is reasonable and intuitive.
2) The table is drawn correctly and the text is written correctly.

IMAGE_1

Figure 7: A success case where the Q-Mirror agent correctly transforms a text-heavy chemistry problem into a structured visual table. The visual representation is accurate, the textual formulation remains fluent, and the modalities exhibit strong synergy, collectively preserving information consistency.

on Q-Mirror-Expert. This confirms that factual grounding in the visual modality is the dominant bottleneck in current one-shot generation approaches.

**Findings for MMQA Evaluation.** Analysis of model-as-judge performance reveals three consistent trends. First, judge accuracy is stratified: Grok-4 achieves the highest agreement with human annotations (62.8%), while weaker models remain below 55%. Second, evaluation difficulty varies by abstraction level. All judges align more closely with humans on Q-Mirror-Grad than on Q-Mirror-Expert. For example, Claude-Sonnet-4 improves from 58.98% on Expert to 71.00% on Grad, reflecting that graduate-level tasks admit more objective criteria, whereas expert-level tasks involve inherently subjective reasoning. Third, correlations among major quality dimensions (IC, CM, QT) are low, validating the non-redundancy of the rubric design and affirming the effectiveness of the underlying evaluation principles. These findings underscore the risk of relying on a single judge and provide motivation for adopting an ensemble of high-performing models to obtain more stable and unbiased evaluation.

**Q-Mirror Agent Performance Improvement.** The Q-Mirror agent yields substantial improvements in MMQA quality. Across all 440 scientific questions, the overall score rises by $\bar{\Delta} = 6.32$ points (from 78.90 to 85.22). More importantly, the pass rate increases from 72% to 95%. This demonstrates that the agent effectively identifies and corrects flaws in nearly one-third of initially suboptimal generations. In side-by-side comparisons, human evaluators prefer agent-refined outputs 78% of the time. Notably, gains are most pronounced on the Expert subset, confirming that iterative refinement is particularly valuable for complex, less standardized scientific problems.

*Original:* A common approximation made in many-body nuclear calculations of some interaction (for example an electroweak interaction) is to suppose that nucleons in a nucleus behave in the same way as free nucleons· Which of the following four assumptions jointly imply this "impulse" approximation? 1· The interaction current only interacts with individual nucleons· 2· The nucleus is transparent apart from the selected nucleon· 3· The quarks internal to the selected nucleon are non-relativistic· 4· The interaction proceeds as if the selected nucleon experiences no binding forces·

*Modified:* Refer to [IMAGE_1]· Based on the visualized assumptions about the interaction of a current with nucleons in a nucleus, which combination of the four labeled assumptions jointly imply the 'impulse' approximation? A) 1,3,4 B) 1,2,4 C) 2,3,4 D) 1,2,3

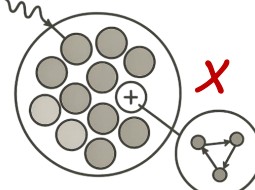

IMAGE_1

*Answer:* B

*Options:* A: 1,3,4  B: 1,2,4  C: 2,3,4  D: 1,2,3

*Error Analysis:* 1) Placing the options after the adapted question adds redundant information (which is redundant with the original options)· 2) Improper image generation

Figure 8: A failure case involving abstract nuclear physics assumptions. The agent attempts to visualize four principles (e.g., 'transparent nucleus'), but produces a generic, ambiguous diagram. The image lacks semantic clarity, offers little complementary information, and results in redundancy with the text.

*Original:* Compounds that have the same molecular formula but are different in their structural arrangement are known as isomers· Isomers have two types, constitutional isomers and stereoisomers· Constitutional isomers have the same molecular formula but differ in their structures· In stereoisomers, molecules are connected in the same way, but their arrangements in space are different· Which of the following organic moieties show optical isomerism? 1· dimethyl 6,6'-dinitro-[1,1'-biphenyl]-2,2'-dicarboxylate 2· methyl 2-hydroxypropanoate 3· benzophenone 4· dimethyl fumarate·

*Modified:* Compounds that have the same molecular formula but are different in their structural arrangement are known as isomers· Isomers have two types, constitutional isomers and stereoisomers· Constitutional isomers have the same molecular formula but differ in their structures· In stereoisomers, molecules are connected in the same way, but their arrangements in space are different· Which of the following organic moieties, as shown in [IMAGE_1], show optical isomerism?

*Answer:* D

*Options:*
A: 3 and 4
B: 2 and 3
C: 1, 2 and 4
D: 1 and 2

IMAGE_1

*Error Analysis:* 1) All four chemical formulas are drawn incorrectly·

Figure 9: A failure case in organic chemistry, where the agent is required to draw molecular structures for four compounds. All structures are chemically incorrect, resulting in factual inconsistency and a breakdown of technical quality.

## F.2 CASE STUDY ANALYSIS OF THE Q-MIRROR AGENT

To further assess the practical performance of the Q-Mirror agent, we present representative case studies spanning both successful and failed generations. These examples provide a nuanced view of the agent's capabilities and limitations, evaluated through our quality rubric (IC, CM, and QT).

**Original:** *An electron is moving along the horizontal axis from the left with a Lorentz factor of 4. A positron is also moving along the horizontal axis, but from the right with a Lorentz factor of 2. After colliding, both annihilate producing 2 photons. The energies of both photons were measured and they have the same value. The directions were also measured. One of the photons was moving in the upper-right direction, while another in the lower-right direction. What is the angle between the directions of the photons?*

**Modified:** *Refer to the scenario and diagram in [IMAGE_1]. After the collision and annihilation of the electron and positron, two photons are produced with equal energies, moving in the directions shown. What is the angle between the directions of the photons?*

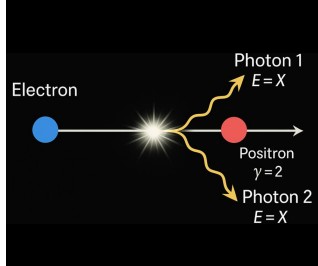

IMAGE_1

**Answer:** *B*

*A: 96   B: 138   C: 74   D: 172*

Figure 10: A success case highlights the transformation of a scientific TQA into its multi-modal counterpart (MMQA). The upper part shows the original TQA, while the lower part displays the O-Mirror generated MMQA, where key textual cues have been converted into a visual representation. The example illustrates how O-Mirror preserves semantic fidelity, introduces a figure to enforce multi-modal dependence, and maintains overall quality and clarity.

In addition to static outcomes, we also examine the iterative dynamics of Q-Mirror's closed-loop pipeline, thereby offering a comprehensive understanding of its operational behavior.

### F.2.1 SUCCESS CASE: STRUCTURING IMPLICIT INFORMATION

**Context.** This case involves a chemistry problem that requires filling in a table, where the table's structure is described only in text (Figure 7).

- **Transformation:** The Q-Mirror agent parses the textual description of rows and columns and successfully converts the implicit structure into an explicit, well-formatted table in the generated image, while retaining the original textual problem statement.

- **Evaluation: IC:** All scientific information is faithfully preserved. **CM:** Both modalities contribute indispensably: the text conveys scientific context, while the image encodes structural information, yielding strong synergy. **QT:** The table is clear, unambiguous, and adheres to conventional scientific presentation standards.

This case exemplifies a prototypical success, in which Q-Mirror effectively externalizes structural representations into the visual modality without sacrificing fidelity.

### F.2.2 SUCCESS CASE 2

To complement the Q-Mirror analysis, we provide a detailed case study of the **O-Mirror agent** (Figure 10). Unlike aggregate benchmarks, this example traces the full pipeline of a single transformation from a text-only QA pair (TQA) to a multi-modal QA pair (MMQA).

**Before (TQA).** The question is entirely text-based, requiring readers to infer visual or structural elements implicitly.

**After (MMQA).** O-Mirror automatically extracts latent visual cues, generates a scientifically aligned illustration, and integrates it with the reformulated question.

This case highlights three critical design principles of O-Mirror:

1. **Preservation of Meaning**: No essential information is lost or distorted in the transformation.

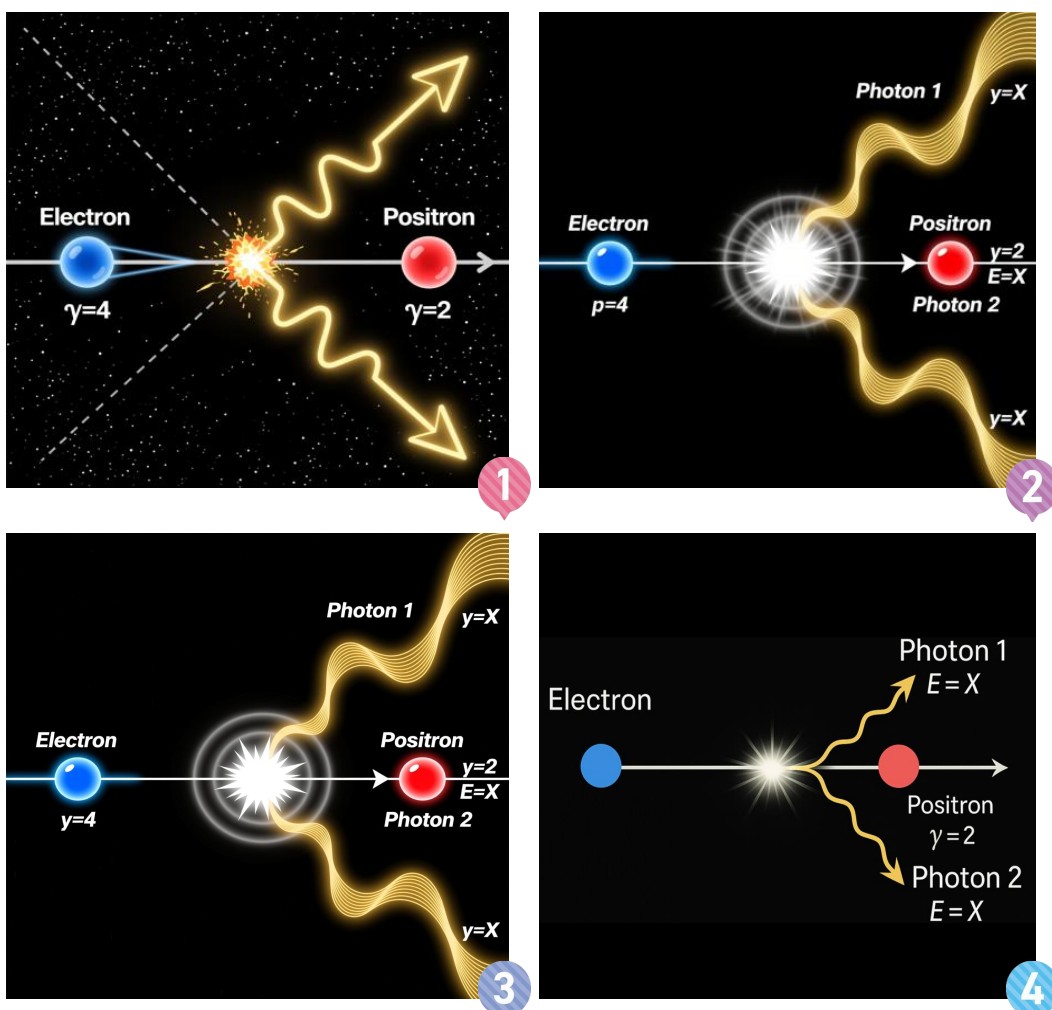

Figure 11: Step-by-step illustration of the Q-Mirror Agent's iterative generation process for the same scientific case shown in Figure 10. (1) **Initial modal conversion:** The original text-only QA pair (TQA) is converted into a visualizable event description, where an electron and a positron collide and produce two photons. (2) **Raw image generation:** A first draft of the multi-modal QA pair (MMQA) is produced, integrating particle labels and energy/momentum annotations, though with incomplete clarity. (3) **Iterative refinement:** Based on evaluator feedback, the image layout and labeling are corrected to enforce physical symmetry and information consistency. (4) **Final polished output:** A simplified and scientifically faithful diagram is generated, achieving clarity, semantic fidelity, and aesthetic quality.

2. **Cross-Modal Integration**: The multimodal question requires reasoning across both text and image, preventing shortcut solutions.

3. **Improved Clarity**: The visual representation enhances comprehension while remaining scientifically plausible.

Through this case study, we demonstrate the interpretability and effectiveness of O-Mirror's pipeline, offering concrete evidence that the framework can faithfully adapt scientific questions into high-quality multi-modal formats.

### F.2.3 FAILURE MODE 1: AMBIGUOUS VISUALIZATION OF ABSTRACT CONCEPTS

**Context.** This case is drawn from a nuclear physics problem concerning the 'impulse approximation', where the agent must visualize four abstract assumptions (Figure 8).

- **Transformation:** The agent produces a generic schematic that attempts to capture multiple abstract principles (e.g., 'the nucleus is transparent' and 'no binding forces'). However, the resulting image fails to distinguish among the assumptions in a meaningful way.

- **Evaluation: IC:** The textual content is preserved, but the visual contributes little additional meaning. **CM:** The lack of discriminative visual elements leads to weak cross-modal integration. **QT:** The image scores poorly on *Semantic Clarity* ($p_{\text{sem}}$), as the intended concepts are not effectively conveyed.

This highlights Q-Mirror's difficulty in translating abstract conceptual descriptions into precise, scientifically interpretable visualizations.

### F.2.4 FAILURE MODE 2: FACTUAL INACCURACY IN TECHNICAL DIAGRAMS

**Context.** This case concerns an organic chemistry problem on optical isomerism, where the agent is tasked with generating molecular structures for four compounds (Figure 9).

- **Transformation:** The agent produces four diagrams intended as molecular structures, all of which are chemically incorrect.

- **Evaluation: IC:** The original entities are not preserved, resulting in a complete violation of information consistency. **CM:** As the diagrams are invalid, the image does not contribute to reasoning; the text alone suffices. **QT:** The diagrams score zero for *Technical Quality* ($p_{\text{tech}}$) due to factual inaccuracies.

This case underscores that factual correctness is non-negotiable in scientific domains and highlights the need for domain-aware verification mechanisms to safeguard against technically invalid outputs.

### F.2.5 ITERATIVE GENERATION DYNAMICS

While the above cases highlight static outcomes, Figure 11 illustrates how the Q-Mirror agent progressively refines a text-only QA pair (TQA) into a high-quality multi-modal QA pair (MMQA).

- **Step (1): Initial conversion** identifies the electron:positron annihilation event but provides only a conceptual representation with limited labeling.

- **Step (2): Raw image generation** introduces particle annotations and energy values, though the layout remains cluttered and lacks physical symmetry.

- **Step (3): Iterative refinement** incorporates evaluator feedback to correct photon emission directions and reduce redundant text, thereby improving information consistency.

- **Step (4): Final output** converges to a polished diagram that balances semantic fidelity, cross-modal integration, and visual clarity.

This example demonstrates the internal dynamics of Q-Mirror's closed-loop generation pipeline, where planner, evaluator, and controller components interact to progressively enhance fidelity, alignment, and presentation quality.

## G  SUCCESS CASE

***Original:*** *Astronomers are studying a system of three exoplanets (Planet1, Planet2, and Planet3) with circular orbits discovered through the TTV method. They have found that the ratio of the equilibrium temperatures between Planet1 and Planet2 is approximately 1.4, and between Planet2 and Planet3, it is about 2.3. They have also found that the ratio of the masses between Planet1 and Planet2 is approximately 1.15, and between Planet2 and Planet3, it is about 1.35. By what factor is the orbital period of Planet3 larger than that of Planet1, if the albedo for all three planets is equal to 0.3 (similar to that of Earth)?*

***Modified:*** *Astronomers are studying a system of three exoplanets (Planet1, Planet2, and Planet3) with circular orbits discovered through the TTV method. The equilibrium temperature ratios and mass ratios between the planets are shown in [IMAGE_1]. By what factor is the orbital period of Planet3 larger than that of Planet1, if the albedo for all three planets is equal to 0.3 (similar to that of Earth)?,*

***Answer:*** *D*
*A: ~ 10.4;*
*B: ~ 4.4;*
*C: ~ 3.2;*
*D: ~ 33.4;*

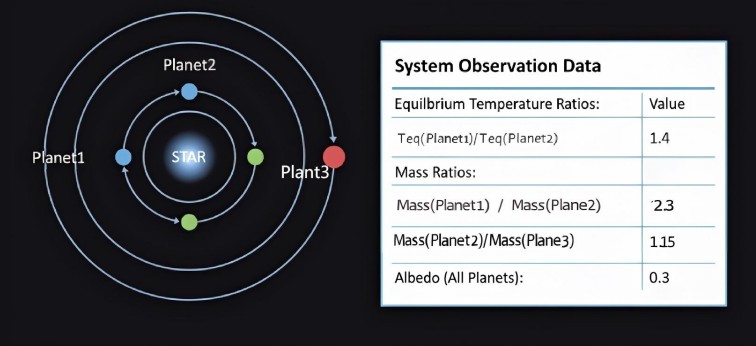

IMAGE_1

***Analysis:***
*1) Enforced Cross-Modal Integration.*
*2) Enhanced Visual Logic.*
*3) Preserved Information Consistency.*

Figure 12: The Final MMQA from a Successful TQA-to-MMQA Transformation. This figure shows the final MMQA generated from a text-only science problem. The visual components exhibit high quality: (a) the system diagram is scientifically plausible, and (b) the data table contains fully legible and correctly rendered text. This example highlights the agent's ability to maintain Information Consistency by preserving all quantitative data, while achieving strong Cross-Modal Integration by making the visual and textual components mutually dependent for solving the problem.

This section provides supplementary case studies to further illustrate the capabilities and internal dynamics of the Q-Mirror agent in transforming text-only questions (TQAs) into high-quality multi-modal questions (MMQAs).

### G.1  CASE STUDY 1: HIGH-FIDELITY ONE-SHOT TRANSFORMATION

Figure 12 showcases the result of a successful one-shot transformation. The source TQA (not shown) consisted of a dense paragraph describing a three-planet system's physical properties. The Q-Mirror agent converted this text into the high-quality MMQA shown in the figure.

The generated MMQA demonstrates excellence across our three evaluation principles:

- **Information Consistency (IC):** All quantitative data from the original text is preserved with perfect fidelity. The ratios for equilibrium temperature (`1.4`) and mass (`2.3`, `1.15`), as well as the shared `Albedo` value (`0.3`), are accurately transcribed into the data table.

- **Cross-Modal Integration (CM):** The resulting MMQA is designed to require reasoning across both modalities. The diagram and table present the necessary scientific data, but the core question is posed in the reformulated text, making it impossible to solve the problem by only consulting one modality.

- **Standalone Quality (QT):** The technical quality of the generated output is high. The planetary system is visualized in a clear, unambiguous diagram. Crucially, the text within the data table is sharp, correctly rendered, and free of the artifacts that can often plague text-to-image models.

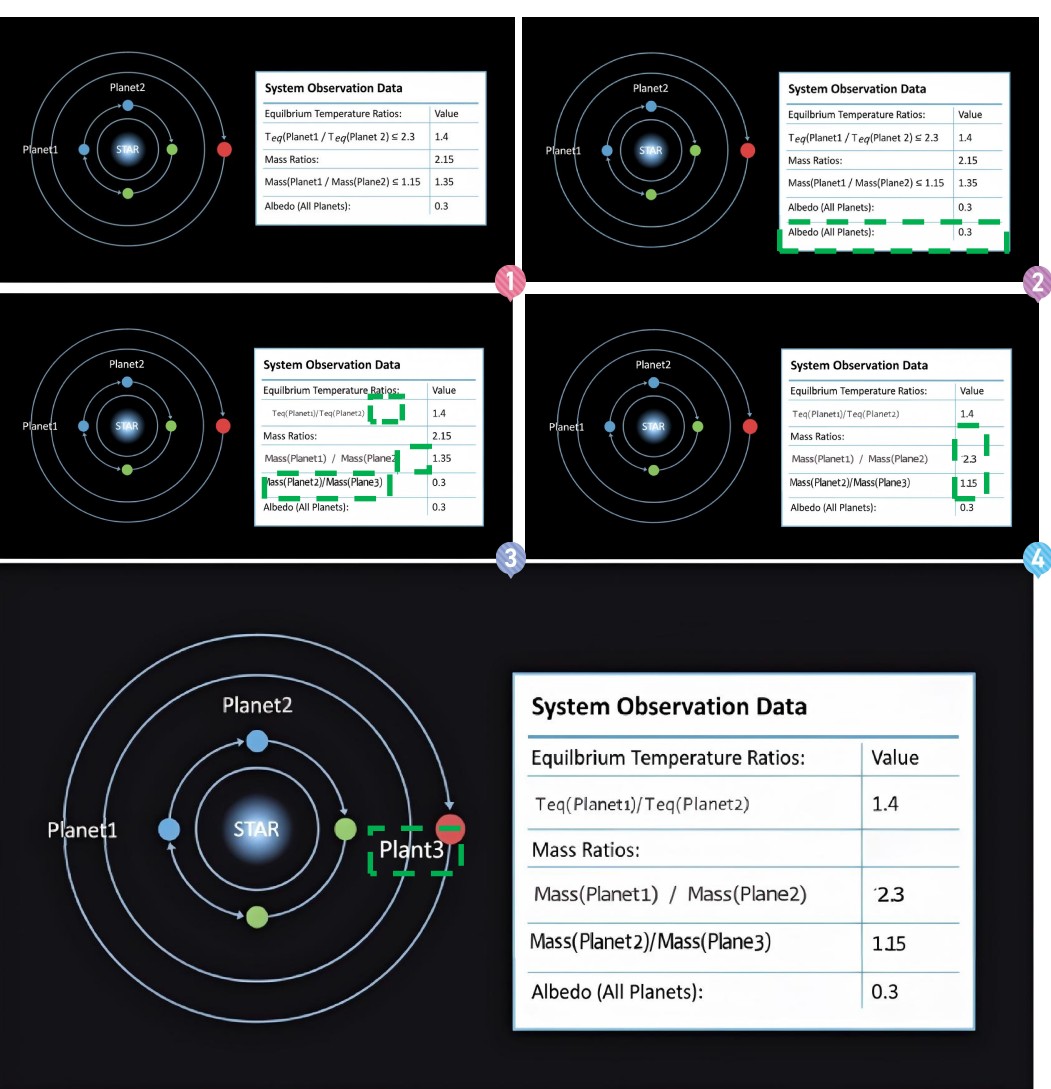

Figure 13: Illustration of the Q-Mirror Agent's Iterative Refinement Workflow. The figure traces the correction of a factual error in a data table across different stages, validating the reliability of the closed-loop refinement process.

This case exemplifies the agent's capacity to generate factually correct and semantically complete multi-modal questions from intricate scientific descriptions.

## G.2 CASE STUDY 2: RELIABLE FACTUAL CORRECTION VIA ITERATIVE REFINEMENT

To illustrate the internal dynamics of our closed-loop pipeline, Figure 13 provides a detailed walk-through of the agent's step-by-step refinement process for a single MMQA.

The process begins with a *(2) Raw Image Generation* that is visually plausible but contains a subtle factual inaccuracy in its data table concerning the planet's `Albedo`. The Evaluator module, leveraging our multi-dimensional quality rubric, identifies this violation of the *Semantic Clarity (SC)* principle.

The structured feedback from the Evaluator is then passed to the *(3) Controller*, which formulates a targeted revision instruction for the Planner. The Planner conditions its subsequent generation attempt on this feedback, producing a corrected version of the MMQA where the data table is now fully consistent with the source TQA. The *(4) Final Polished Output* is therefore not only visually appealing but also factually sound.

*Original:* *Consider an ecosystem where seeds, berries, and grasses function as the foundational producers. These resources support various consumers: grasses provide energy for marmots, elk, and grasshoppers, while berries are consumed by chipmunks, grizzly bears, grasshoppers, and grouse, and seeds are eaten by chipmunks and grouse. The energy flow continues as grouse act as omnivores by preying on grasshoppers in addition to eating plants, while goshawks hunt chipmunks. Meanwhile, grizzly bears function as top predators by feeding on chipmunks, marmots, and elk. Based on the trophic relationships described in this text,* the grasshopper population would most likely increase if:

*Modified:* *From the above food web diagram shown in [image_1],* grasshopper population increase if:

*Answer:* C
A: grouse decrease;
B: chipmunk increases;
C: grasses increases;
D. elk increase;

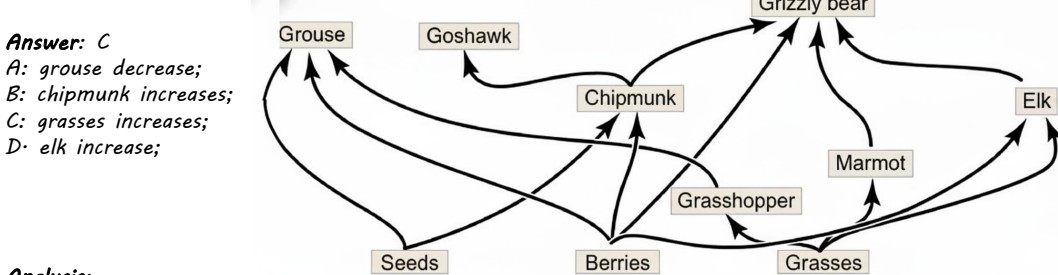

IMAGE_1

*Analysis:*
1) Enforced Cross-Modal Integration.
2) Enhanced Visual Logic.
3) Preserved Information Consistency.

Figure 14: The Final MMQA from a TQA-to-MMQA Transformation in Biology. This figure shows the successful conversion of a text-based ecosystem description into a clear food web diagram, enhancing visual logic and enforcing cross-modal reasoning.

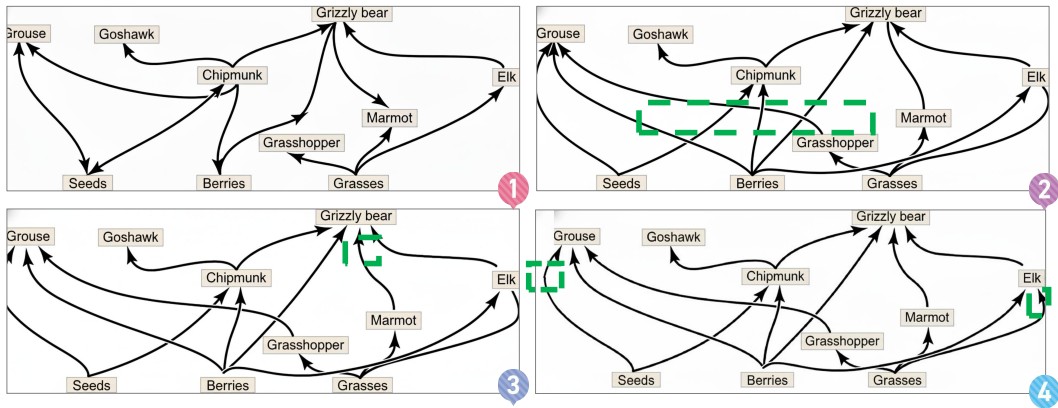

Figure 15: Iterative Refinement of a Food Web Diagram's Layout. This figure shows the four-stage process where the Q-Mirror agent improves the visual clarity of an ecosystem diagram by reorganizing nodes and simplifying arrow paths based on automated feedback, enhancing cognitive efficiency.

This case highlights that the iterative refinement loop is a structured mechanism for improving factual correctness. By identifying and rectifying errors, the agent reliably converges on a high-quality output, demonstrating the robustness of the Q-Mirror framework.

### G.3 CASE STUDY 3: ENHANCING LOGICAL REASONING IN ECOSYSTEM ANALYSIS

Figure 14 presents the successful outcome of transforming a descriptive biology problem into a visual reasoning task. The source TQA (not shown) contained a paragraph detailing the complex trophic relationships within an ecosystem. The Q-Mirror agent successfully externalized these relationships into the intuitive and scientifically accurate food web diagram shown.

The transformation excels in the following ways:

- **Information Consistency (IC):** The agent correctly identifies all producers (seeds, berries, grasses) and consumers. It then accurately maps every predator-prey link described in the source text, such as the omnivorous nature of the grouse and the top predator role of the grizzly bear. No information is lost or altered.

- **Cross-Modal Integration (CM):** The descriptive context from the original text is removed and embedded into the visual diagram. The reformulated question becomes concise and entirely dependent on the diagram, forcing a user to visually trace the food web's connections to determine the impact on the grasshopper population.

- **Standalone Quality (QT):** The key advantage of this MMQA is the enhancement of logical reasoning. The food web diagram provides a holistic representation of the ecosystem, which is more efficient for analyzing systemic effects than parsing a linear text. The final diagram is clear, uncluttered, and adheres to scientific conventions.

This case exemplifies how Q-Mirror can convert dense, descriptive problems into structured visual tasks that are more aligned with real-world scientific analysis.

### G.4 Case Study 4: Visual Layout Refinement for Cognitive Clarity

Figure 15 illustrates the agent's iterative process for refining not just factual accuracy but also the visual clarity of a generated diagram. While the initial generation may be factually correct, its layout can be suboptimal for human reasoning.

The refinement cycle proceeds as follows: The *(1) Raw Generation* produces a food web that correctly represents all trophic links. However, the *(2) Evaluator* identifies a layout issue—specifically, that multiple crossing arrows originating from the "Berries" node create visual clutter, hindering the easy identification of specific food chains.

Based on this feedback, the *(3) Controller* instructs the Planner to "reorganize nodes to improve clarity and minimize intersecting arrows." The Planner then generates improved layouts. The *(4) Final Polished Output* is a version that is semantically identical but visually superior. The spatial arrangement of species is more organized, reducing cognitive load and making the ecosystem's structure more immediately apparent.

This case study demonstrates that the Q-Mirror's refinement loop extends beyond simple factual correction to optimize the cognitive efficiency of the generated visualizations, a crucial aspect for high-quality scientific diagrams.