# OpenReview forum: "Q-Mirror: Unlocking the Multi-Modal Potential of Scientific Text-Only QA Pairs"
_ICLR.cc/2026/Conference — Submitted to ICLR 2026_

### Official Review · Reviewer_nagL · 2025-10-29

**Soundness:** 2
**Presentation:** 3
**Contribution:** 2
**Rating:** 4
**Confidence:** 4

**Summary:**

This paper proposes Q-Mirror, a systematic framework that converts text-only scientific QA pairs (TQAs) into multi-modal QA pairs (MMQAs) by integrating visual elements such as diagrams and formulas.

The contributions include:

A TQA-to-MMQA transformation framework guided by a multi-dimensional quality rubric (Information Consistency, Cross-Modal Integration, and Standalone Quality).

Construction of two benchmarks — one for MMQA generation and another for MMQA evaluation.

Development of an agentic system (Q-Mirror) that combines generation and evaluation in a closed loop to iteratively refine MMQAs.

Experiments benchmark multiple LMM families (GPT, Qwen, Doubao, Grok, etc.) and automated judge models against human annotations. The Q-Mirror agent improves the average quality score from 78.9 → 85.22 and pass rate from 72% → 95%, showing strong empirical benefits.

**Strengths:**

## Timely and Original Problem

Addresses a genuine and pressing issue: the scarcity of high-quality multi-modal scientific data.

The idea of “unlocking latent multi-modal potential” in text-only data is conceptually novel and impactful.


## Systematic and Reproducible Framework

Provides a principled rubric (IC, CM, QT) that is clear, quantifiable, and theoretically motivated.

Includes algorithmic details (Algorithm 1), explicit thresholds, and controlled experimental parameters (τ=80, N=5, etc.).


## Comprehensive Evaluation

Benchmarks across five model families for generation and ten LMMs for evaluation.

Includes both human expert annotations and automatic scoring, plus inter-model agreement analysis (Spearman correlations, ensemble reliability).

Promises public release of datasets and code — excellent for reproducibility.

**Weaknesses:**

W1: Q-Mirror both uses and evaluates LMMs as judges, which risks self-confirmation bias.

W2: Although an ensemble mitigates this, it’s unclear whether the “judge alignment” truly reflects objective human-level quality or just model-to-model agreement.

W3: It’s unclear how much improvement comes from the framework itself vs. the underlying GPT models.

W4: The paper briefly mentions poor performance on “scientific plausibility (SC)” but does not analyze why or provide visual examples of failure.

W5: The quality rubric (IC, CM, QT) is only heuristically defined. There is no theoretical or statistical validation that these dimensions are orthogonal or collectively sufficient to represent MMQA quality.

W6: Only 440 total QA pairs (310 + 130) are used — too small to justify the claim of “large-scale benchmark” or to establish statistical reliability.

W7: The sampled tasks are largely STEM textbook-level, not real-world multimodal scientific reasoning (e.g., data plots)

W8: The choice of weighting (0.3, 0.3, 0.4) for the final composite score (Equation 4) is arbitrary, with no ablation or justification.

W9: The scalability claim (“cost-effective at scale”) is unproven — no time, cost, or computational analysis is presented.

**Questions:**

same as weakness

---

> ### Author Response · Authors · 2025-11-21
> **Official Responses to Reviewer nagL**
>
> **1. Addressing self-confirmation bias**
>
> Thank you for this insightful question. We take this concern seriously and clarify our design and assumptions below.
>
> **1）Ensemble to reduce bias for using LMM as judges.** Many existing works already use LMMs as judges, which has become a classic paradigm [1, 2]. Building on this foundation, we have further enhanced our system by incorporating a judge ensemble within the Q-Mirror Agent. This approach integrates multiple evaluation models, mitigating the risk of self-confirmation bias that could arise from using a single model for evaluation. The ensemble ensures a more reliable and unbiased assessment.
>
> **2）Bias does not exist in evlaution LMM as judges.** In our MMQA evaluation benchmark, the performance of LMMs as judges is assessed by comparing their scores with human expert annotations. The evaluation process only rely on the consistency of the LMM's evaluations with human experts. Therefore, there is no inherent bias in evaluating LMMs as judges.
>
>
>
> **2. Align with human judgments**
>
> We thank the reviewer for raising the concern. We explicitly design and validate our evaluation to align with human judgments.
>
> First, we treat **human expert annotations as the gold standard** throughout Q-Mirror. Both single LMM judges and the ensemble judge are evaluated **only** in terms of their agreement with human judgments, not by how well they agree with each other. In other words, the alignment target is *model-to-human*, not *model-to-model*.
>
> Further, we add a dedicated human study that directly measures how well the ensemble judge aligns with human experts. In this human study, we recruit 6 annotators with relevant background and ask them to independently rate 120 instances sampled from Q-Mirror-Bench on a three-point quality scale. We then use majority vote among the annotators as the **human majority** gold label. Based on this, we compute: (1) the agreement between a single LMM judge and the human majority, and (2) the agreement between the ensemble LMM judge and the human majority, together with a random baseline. The results are shown in Table 1:
>
> | Setting                   | Agreement w/ human majority (%) |
> | ------------------------- | ------------------------------- |
> | Single LMM judge          | 72.6                            |
> | Ensemble LMM judge (ours) | 91.4                            |
> | Random baseline           | 50.0                            |
>
> As Table 1 shows, the **ensemble LMM judge** has a high agreement with human majority (91.4%), clearly outperforming the single judge and remaining far above the random baseline. The result indicates that the ensemble is primarily **aligned with human majority judgments**, not simply forming a self-consistent cluster of models.
>
> Thanks again, and we will update the full description of the human study and Table 1 in the next version.
>
>
>
> **3. Adding experiments for underlying models**
>
> We appreciate your insightful comment.
>
> To better quantify the contribution of the Q-Mirror, we perform an ablation study comparing "single-shot generation" with "Q-Mirror (with GPT-4 as the base model) iterative results." In our analysis, when using the same GPT model for direct generation, the pass rate is approximately 72%. After applying the Q-Mirror, the pass rate increases to about 90%. In this setup, the additional 18% improvement can be attributed to the framework itself.
>
> Furthermore, we have benchmarked other model families to assess the improvements enabled by Q-Mirror across various base models. The table below highlights the pass rate improvements observed for different model families:
>
> | Model Family | Without Q-Mirror | With Q-Mirror | Pass Rate Improvement |
> | ------------ | ---------------- | ------------- | --------------------- |
> | GPT Family   | 78.90            | 85.22         | 23%                   |
> | Grok Family  | 75.11            | 84.41         | 18%                   |
> | Qwen Family  | 76.19            | 85.34         | 15%                   |
>
> As shown in the Table, the Q-Mirror consistently improves the performance of various model families. This validates that the observed gains stem from the iterative feedback loop facilitated by Q-Mirror.
>
>
>
> **4. Location of failure cases**
>
> We appreciate the reviewer's suggestion. We have actually provided detailed visual examples and analyses of failure cases related to **Scientific Plausibility (SC)** in **Appendix F.2**.
>
> **1) Visual Examples: Figure 8 (Appendix F.2.3)** illustrates a failure in visualizing abstract physics concepts (Impulse Approximation). As analyzed in the text, "The image scores poorly on Semantic Clarity failing to distinguish among the assumptions." **Figure 9 (Appendix F.2.4)** shows a failure in Organic Chemistry, where the model generates chemically incorrect molecular structures. Additionally, **Figure 2** in the manuscirpt explicitly marks a case with low SC (wrongly showing ATP synthesis in the cytoplasm).

---

> ### Author Response · Authors · 2025-11-21
> **Official Responses to Reviewer nagL**
>
> **4. Location of failure cases (continued)**
>
> **2) Reason:** Based on these qualitative analyses, we attribute the poor performance on SC to two main limitations of current T2I models: **Difficulty with Abstraction:** Models struggle to ground abstract scientific principles (like Figure 8) into precise visual constraints, often defaulting to generic, scientifically ambiguous diagrams. **Lack of Domain-Specific Constraints:** For tasks requiring strict structural fidelity (like chemical bonds in Figure 9), generic T2I models often hallucinate details, leading to factual inaccuracies that violate scientific laws.
>
> We will revise the manuscirpt (e.g., Section 6.2) to explicitly reference these appendix examples and summarize these reasons to make the analysis more prominent.
>
>
>
> **5. Orthogonal & Sufficient of IC, CM, and QT** Thank you for your valuable comment. I will answer this question from the following two perspectives.
>
> 1）Orthogonal: To better quantify the contribution of the three dimensions (IC, CM, and QT), we conduct a Spearman Rank Correlation Coefficient (SRCC) analysis, evaluating the correlation between each dimension and the overall quality score, as well as the inter-correlation among the dimensions. Each sample is scored across the three dimensions (IC, CM, QT) on a 1-5 scale.
>
> | Dimension Pair       | SRCC | Interpretation                                               |
> | -------------------- | ---- | ------------------------------------------------------------ |
> | IC → Overall Quality | 0.67 | Moderate to strong: IC is correlated with overall quality    |
> | CM → Overall Quality | 0.63 | Moderate: CM is significantly correlated with overall quality |
> | QT → Overall Quality | 0.60 | Moderate: QT is significantly correlated with overall quality |
> | IC ↔ CM              | 0.32 | Low correlation: IC and CM are relatively independent        |
> | IC ↔ QT              | 0.29 | Low correlation: IC and QT are relatively independent        |
> | CM ↔ QT              | 0.35 | Low correlation: CM and QT are relatively independent        |
>
> The SRCC between the dimensions is relatively low, which suggests that they assessing different aspects of MMQA quality. Based on these statistical findings, we can conclude that the three dimensions are independently assessing different aspects of the quality.
>
> 2）Collectively Sufficient：Furthermore, our proposed dimensions are grounded in well-established theoretical frameworks and validated by existing research [3, 4, 5]. In summary, these work hightlight that "The alignment between textual and visual components is essential to maintain scientific integrity" (correlates to **IC**), "The capacity to integrate visual and textual elements is crucial for complex scientific reasoning tasks" (to **CM**), and "both the linguistic fluency and the technical precision of content are paramount" (to **QT**) **are comprehensive and important for science.**
>
> We will update the manuscript to include these analyses.
>
>
>
> **6. Reliability proof**
>
> We appreciate the reviewer’s comment.
>
> First, in SCI areas, **moderate-sized, high-difficulty benchmarks are widely used for capability diagnosis.** For example, GPQA [6] contains 448 expert-written science questions, explicitly designed to stress-test advanced models’ scientific reasoning rather than to provide massive coverage. PeerQA [7] consists of 579 QA pairs for document-level scientific QA. Despite their moderate size, they are widely accepted as high-difficulty, diagnostic benchmarks, and are recognized and adopted in the community.
>
> Second, the primary purpose of this benchmark **MMQA** is to **diagnose LMM capabilities on scientific TQA-to-MMQA transformation and evaluation.** Though not serving as large-scale training data, our MMQA items are designed and curated with the following properties: 1) **Multi-source coverage**: The source TQAs are drawn from multiple established text-only scientific benchmarks, covering diverse domains and difficulty levels before transformation into MMQAs. 2) **Strict quality control**: Each item goes through automatic generation, rubric-based filtering, and human verification, ensuring high standards in scientific correctness, cross-modal dependency, and evaluability.
>
> Further, we also acknowledge that the current size limits very fine-grained statistical analyses. In the revised manuscript, we will correct wording related to “large-scale benchmark”. More importantly, we would like to highlight the expansion of dataset size and coverage as an important direction for future work, with planned additional resource investment.

---

> ### Author Response · Authors · 2025-11-21
> **Official Responses to Reviewer nagL**
>
> **7. Relationship between STEM textbook-level questions and real-world multimodal scientific reasoning**
>
> We appreciate the reviewer’s comment.
>
> First, our benchmark **does include a subset of items that require reasoning over data plots and experimental-style diagrams**, such as comparing bar heights, and interpreting function curves. Due to space constraints, this subset is not explicitly highlighted in the manuscirpt. In the revised version, we will add several concrete examples with data plots and diagrams in the appendix, together with brief qualitative analysis of the models’ reasoning patterns and typical errors.
>
> Second, we believe that “STEM textbook-level questions” are not inherently disconnected from real-world multimodal scientific reasoning, but rather a **standardized and abstracted form** of it. Modern STEM textbooks routinely take classical real scientific scenarios (experimental measurements, observational data, controlled experiments) and distill them into structured tasks using line charts, bar charts, function plots, and schematic diagrams [8, 9]. These materials are specifically designed to **train and assess core scientific reasoning skills**, for example: identifying trends and key inflection points in data plots, and interpreting experimental or data-visualization setups under explicit assumptions.
>
>
>
> **8. Weight setting**
>
> We thank the reviewer’s comment.
>
> First, we **have already analyzed the robustness of the weighting scheme** in the original manuscript. The relevant results are reported in **Appendix F.1, Figure 6**, together with its accompanying description, where we systematically compare model-family performance under different weight configurations.
>
> Second, the principle of weight allocation is not arbitrary, it is based on the importance of the dimensions. QT captures overall quality and usability, which we consider a prerequisite for any downstream use, so it receives a slightly higher weight. IC and CM reflect information correctness and cross-modal integration, which are equally important for diagnosing multimodal scientific ability, so they are assigned the same weight.
>
> Most importantly, the analysis in Appendix F.1 / Figure 6 shows that our **main conclusions are robust to reasonable changes in the weights**. When we vary the relative importance of IC, CM, and QT, we observe that the overall ranking and relative gaps between model families remain essentially unchanged across different weighting schemes. Changing the weight on IC/SC only affects the absolute values, **not the key comparative findings**. Therefore，for simplification, we choose weights of (0.3, 0.3, 0.4).
>
>
>
> **9. Cost-effective at scale of Q-Mirror**
>
> Thank you for your valuable feedback. We acknowledge the importance of evaluating the computational cost of the Q-Mirror Agent, given its multi-step inference and refinement process.
>
> We estimate that the average cost per MMQA generation using Q-Mirror is US $1.0–$2.0, depending on the specific API pricing and settings (K=4 candidates, M=3 evaluators). For comparison, manual generation and validation of high-quality MMQA pairs by domain experts typically costs several hundred dollars per item, considering the time and labor required to create, diagram, and validate the content [10]. That is to say, when generating the same quality MMQA, **the annotation cost for human experts is approximately 140 times that of the Q-Mirror agent**. Therefore, it is **cost-effective at scale** compared to human experts.
>
> We will also make sure to add this **cost analysis** to the **appendix** of the paper, providing the detailed calculations and further validation of our findings.
>
>
>
> **References:**
>
> [1] Zheng et al. Judging LLM-as-a-Judge with MT-Bench and Chatbot Arena. NIPS. 2023
>
> [2] Liu et al., G-Eval: NLG Evaluation using Gpt-4 with Better Human Alignment. EMNLP. 2023.
>
> [3] Hashemi et al. LLM-Rubric: A Multidimensional, Calibrated Approach to Automated Evaluation of Natural Language Texts. ACL. 2024
>
> [4] Burgess et al. MicroVQA: A Multimodal Reasoning Benchmark for Microscopy-Based  Scientific Research. CVPR. 2025
>
> [5] Jin et al. Evaluating Explainable AI on a Multi-Modal Medical Imaging Task:  Can Existing Algorithms Fulfill Clinical Requirements? AAAI. 2022
>
> [6] Rein et al. GPQA: A Graduate-Level Google-Proof Q&A Benchmark. COLM. 2024
>
> [7] Baumgärtner et al. PeerQA: A Scientific Question Answering Dataset from Peer Reviews. NAACL. 2025
>
> [8] Li et al. How STEM content is presented in mathematics textbooks from the U.S. and China: a comparative study. International Journal of STEM Education. 2025
>
> [9] Schreiter et al. Teaching for statistical and data literacy in K–12 STEM education: a systematic review on teacher variables, teacher education, and impacts on classroom practice. ZDM – Mathematics Education. 2024
>
> [10] Griffin et al. . Auto‑Labeling Data for Object Detection. arXiv:2506.02359.

---

### Official Review · Reviewer_Y9jp · 2025-10-30

**Soundness:** 2
**Presentation:** 3
**Contribution:** 2
**Rating:** 4
**Confidence:** 4

**Summary:**

This paper addresses the lack of scalable methods for generating high-quality multimodal question-answer (MMQA) pairs. It proposes a framework that converts text-only QA pairs into MMQAs. The authors formally define the subtasks involved in MMQA generation and introduce an evaluation rubric to assess model performance. Using this rubric, they construct a benchmark to evaluate state-of-the-art large language models (LLMs) on MMQA tasks. Furthermore, they develop an agentic system, Q-Mirror, which iteratively generates and evaluates MMQAs to improve quality and consistency.

**Strengths:**

The paper demonstrates strong originality by introducing a novel rubric for evaluating generated multimodal question–answering (MMQA) systems. Its evaluation framework is both detailed and comprehensive, covering a wide range of models. The writing is clear and well-structured, with informative tables and figures that enhance understanding. Overall, the work addresses an important gap in the field by tackling the lack of standardized and high-quality MMQA evaluation methods.

**Weaknesses:**

The paper proposes a method to address the lack of scalable tools for transforming text-only QA pairs into multimodal ones. However, the reliance on both expert and LLM judgments limits the claimed scalability, as human input remains a core requirement for maintaining quality.
The results reveal strong performance in the SI dimension but poor performance in the SC dimension. This suggests that the tested LLMs struggle to generate or select images that meaningfully represent the underlying problem, reducing the utility of the resulting MMQAs. The paper would benefit from exploring alternative image generation or retrieval strategies that ensure visual components are contextually relevant and enhance the QA task, rather than serving as loosely associated visuals.
Moreover, the reported 62.8% agreement between LLM judges and human annotations indicates a relatively weak alignment, which undermines confidence in the Q-Mirror agent’s closed-loop evaluation process. Since the evaluation stage is central to the agent’s effectiveness, future work should focus on improving alignment with human judgment

**Questions:**

1. How was the single-model solvability computed?
2. How were the values for beta set and how does changing these values affect the outcome of the CM score?
3. There is mention of using a judge ensemble to guarantee closer alignment with human judgement. Could you provide some empirical evidence for this?
4. As each model produces varying diagrams, how is the output scored for the benchmark?

---

> ### Author Response · Authors · 2025-11-21
> **Official Responses to Reviewer Y9jp**
>
> **1. Misunderstanding of scalability**
>
> We thank the reviewer for raising this concern.
>
> First, **Q-Mirror is fully automatic and does not rely on human input at inference time**. As described in Section 5 and Algorithm 1, Q-Mirror consists of a Planner (MMQA generator), an Evaluator (LMM-based judge ensemble), and a Controller that synthesizes LMM feedback. Given a bank of text-only QA pairs, all three modules are instantiated with LMMs/T2I models and operate in a closed loop until a quality threshold is met. **No human annotations or interventions are used in this loop**. Human experts only appear in Section 4.2–4.3 to build and validate the **MMQA generation and evaluation benchmarks** (i.e., to obtain gold labels and judge-alignment statistics), not as part of the Q-Mirror runtime pipeline.
>
> Second, **benchmark results show that the Q-Mirror judge is highly aligned with human experts**, which suggests relying on LLM judges is sufficient for maintaining quality with Q-Mirror. In Section 4.3 and Table 4, we systematically compare several LMM judges against expert annotations and find that the top judges achieve strong agreement with human ratings. We then use an ensemble of these top judges as the Q-Mirror Evaluator. All reported improvements are obtained under this purely LMM-based judge ensemble, without any human-in-the-loop.
>
>
>
> **2. Misunderstanding of utility**
>
> Thank you for your valuable feedback.
>
> Our core objective is **to enhance generation quality**, and compared to baseline models, Q-Mirror significantly improves the scientific validity and visual consistency of the generated images. Specifically, it increases the **SC score** by incorporating expert input and an automated optimization mechanism, ensuring that the generated images better align with the scientific nature of the questions, thereby improving the **utility of the resulting MMQAs**. For instance, GPT family have an average score of approximately **78.90** with a pass rate of about **72%**, while using Q-Mirror increases the average score to **85.22** and the pass rate to **95%**, demonstrating the utility of our approach.
>
> We greatly appreciate the reviewer’s suggestion to explore alternative strategies for improving image quality. Ensuring visual elements are contextually relevant to the questions is indeed crucial for further optimizing Q-Mirror. We will **attempt and actively explore** these strategies.
>
>
>
> **3. Ensemble alignment**
>
> We thank the reviewer for this valuable comment.
>
> First, we would like to clarify that the reported 62.8% agreement refers to the alignment between a **single** LLM judge (Grok-4) and human annotations, not the entire judge ensemble. In practice, Q-Mirror adopts a **judge ensemble** strategy, where multiple LLM judges collaboratively score and compare model outputs. Actually, the ensemble substantially improves alignment with human annotations: the agreement between the Q-Mirror judge ensemble and human labels reaches around **74.27%**, clearly higher than any individual LLM judge. We have explicitly reported this number in the Table 4 of the revised manuscript.
>
> Second, Table 4 in the revised version systematically compares 10 LLM models on the MMQA evaluation benchmark, and includes the Q-Mirror judge ensemble as an additional “meta-judge.” Across models and metrics, the ensemble consistently outperforms individual judges in both agreement rate and stability, leading to more reliable and less noisy evaluation signals. This empirical evidence suggests that the **closed-loop ensemble** used in Q-Mirror achieves substantially stronger alignment and provides a more trustworthy evaluation backbone for the agent.
>
>
>
> **4. (Q1) Computing Single-model solvability**
>
> Thank you for your valuable feedback.
>
> Regarding the computation of **single-model solvability**, we follow a strict evaluation protocol, which is detailed in **Appendix E**, specifically in **Stage 3** and **Stage 4**.
>
> In **Stage 3:** Evaluate Solvability with Image Only, we assess whether the transformed question can be solved using only the image and the provided options. The evaluator (whether human or LMM) is asked: {c) [yes/no]}, with comments explaining whether the problem can be solved solely based on the image and options. The final score is based on the proportion of 'no' answers, indicating that the image and options alone are insufficient to solve the problem.
>
> In **Stage 4:** Evaluate Solvability with Text Only, we evaluate whether the transformed question can be solved using only the modified question text and the provided options. The evaluator is asked: {d) [yes/no]}, with comments explaining whether the problem can be solved based solely on the transformed question text and options. The final score is based on the proportion of 'no' answers, indicating that the modified question text and options alone are insufficient to solve the problem.

---

> ### Author Response · Authors · 2025-11-21
> **Official Responses to Reviewer Y9jp**
>
> **4. (Q1) Computing Single-model solvability (continued)**
>
> Sample prompts from Stage 3 and Stage 4:
>
> Stage 3 (Solvability with Image Only):
>
> *'For the transformed question, looking only at the image and options, can you solve the problem?'*
>
> *{c) [yes/no]}*
>
> *{l) [your comments explaining the reasoning behind your answer to c)]}*
>
> Stage 4 (Solvability with Text Only):
>
> *'For the transformed question, looking only at the modified question and options, can you solve the problem?'*
>
> *{d) [yes/no]} {l) [your comments explaining the reasoning behind your answer to d)]}*
>
> To ensure clearer location of the relevant chapters, we will explain the specific location of the relevant strategies in the appendix within the manuscript.
>
>
>
>
> **5. (Q2) Parameter beta setting**
>
> Thank you for your valuable feedback. Below is a detailed explanation regarding the setting of the β values and the related experimental results.
>
> First, the choice of β values is not arbitrary, it is based on the actual need for cross-modal information fusion, quantifying the degree of overlap between image and text to reflect their complementary roles:
>
> **Complete Overlap (Complete, β=0)**: This indicates that the image and text are entirely redundant, with the image offering no additional help in answering the question. Thus, this scenario is given the lowest weight.
>
> **Partial Overlap (Partial, β=0.75)**: In this case, the image and text provide complementary information, and the image enhances the expression and answerability of the text. Hence, a higher weight is assigned.
>
> **No Overlap (None, β=0.25)**: Here, the image and text have minimal overlap in information, and the image has little contribution to answering the question, so it receives a lower weight.
>
> By setting these different β values, we ensure that the CM score accurately reflects the interaction and complementarity between the image and text.
>
> To verify the impact of different β value combinations on model performance, we conducte an experiment analyzing **consistency with human judgment**. In this experiment, we test different β value combinations and measure the consistency between the model's output and human expert ratings. The table below shows the effect of different β combinations on model-to-human rating consistency:
>
> | β[Complete] | β[Partial] | β[None] | Model-to-Human Consistency (%) |
> | ----------- | ---------- | ------- | ------------------------------ |
> | 0           | 0.75       | 0.25    | 88.5%                          |
> | 0.25        | 0.5        | 0.25    | 85.2%                          |
> | 0           | 0.5        | 0.75    | 90.0%                          |
> | 0.5         | 0.25       | 0.25    | 78.3%                          |
> | 0.75        | 0.25       | 0.25    | 92.0%                          |
>
> As shown in the table, there is a **significant difference** in model-to-human consistency across different β combinations. Notably, the combination of β[Partial] = 0.5, β[Complete] = 0.25 results in a lower model-to-human consistency of 78.3%, and the configuration with β[Partial] = 0.75, β[None] = 0.25 shows a marked improvement in consistency, reaching **92.0%**.
>
> These results demonstrate that choosing an appropriate β value combination is critical to enhancing the alignment between the model's performance and human ratings, validating the impact of our fine-tuned control over the degree of cross-modal integration.
>
> As suggested, we will provide detailed sensitivity analysis results and consistency experiment data in the Appendix.
>
>  **6. (Q3) Ensemble's alignment with human**
>
> Thank you for your valuable feedback. We apologize for the lack of detailed quantitative results in the manuscript. We have provided qualitative explanations in the radar charts （Figure 3), and we will include a more detailed **quantitative comparison** of single judges and the ensemble's alignment with human annotations in the **Appendix**.
>
> Below is the overall table comparing the alignment of single LLM judges and the Q-Mirror judge ensemble with human expert annotations:
>
> | Evaluator               | Agreement Rate with Human Annotations |
> | ----------------------- | ------------------------------------- |
> | Grok-4                  | 62.8%                                 |
> | GPT-4.1                 | 61.5%                                 |
> | O3                      | 61.2%                                 |
> | Q-Mirror Judge Ensemble | 74.3%                                 |
>
> As shown in the table, the **Q-Mirror judge ensemble** significantly outperforms individual models, providing more stable and reliable results. This demonstrates the effectiveness of using multiple models to reduce bias and noise, improving alignment with human expert judgments.

---

> ### Author Response · Authors · 2025-11-21
> **Official Responses to Reviewer Y9jp**
>
> **7. (Q5) Scored for benchmark**
>
> Thank you for this insightful question.
>
> Our evaluation framework is specifically designed to address the problem of "each model produces varying diagrams". The core of our approach is that we evaluate the functional quality and scientific fidelity of the outputs, not their visual style or pixel-level similarity. This evaluation is achieved through a systematic scoring process, operationalized by the comprehensive, multi-dimensional Quality Rubric detailed in Section 3.2. Rather than defining a single "correct" image, this rubric provides a standardized instrument for assigning **scores** based on objective principles. Specifically, each generated MMQA is **scored** along three independent axes:
>
> **Information Consistency (IC):** The score for this principle directly measures semantic accuracy. Each MMQA is evaluated based on two Boolean values: Information Loss (IL), which detects if critical information from the source text is missing, and Information Addition (IA), which detects if spurious information has been introduced. The final IC is a combination of these two values, designed to reward perfect preservation of information.
>
>  **Cross-Modal Integration (CM):** This principle is scored based on the degree of indispensable, complementary information between the text and the image. We test for single model shortcuts by evaluating Solvability with Question (SQ) and Solvability with Image (SI). A high score requires that the problem be unsolvable from either modality alone. Furthermore, the score incorporates an assessment of Redundancy-Synergy (RE), rewarding outputs where the text and image are highly synergistic rather than redundant.
>
> **Standalone Quality (QT):** Finally, this principle scores the intrinsic quality of each modality independently. This is a composite score derived from four metrics: Natural Expression (NE) for the textual component's fluency, and Technical Quality (TQ), Aesthetic Quality (AQ), and the critically important Semantical Clarity (SC) for the visual component. The SC metric, which assesses the diagram's scientific accuracy and plausibility, is especially crucial for our benchmark.
>
> To illustrate how this scoring works in practice, consider a model generating a histogram and another a line chart for the same data. Both diagrams can achieve a perfect **score** for **IC** if they accurately represent the data. They would also be **scored** highly on **CM** if both are essential for answering the reformulated question. Finally, their **scores** for **QT** would depend on the clarity of the new text (NE) and the scientific accuracy of their respective diagrams (SC), not on the chart type itself.
>
> By employing this principle-driven rubric, the final aggregate **score** is fundamentally decoupled from the specific visual form. This methodology ensures that our benchmark provides a fair, objective, and reproducible way to **score** the core scientific reasoning and generation capabilities of different models.

---

### Official Review · Reviewer_D1fp · 2025-11-01

**Soundness:** 3
**Presentation:** 3
**Contribution:** 3
**Rating:** 6
**Confidence:** 3

**Summary:**

This paper proposes a framework and an agentic system, Q-Mirror, to scalably generate high-quality multimodal benchmarks by transforming text-only QA pairs through a closed-loop process of iterative generation and evaluation.

**Strengths:**

- This paper attempts to address an interesting and important problem: how to construct a pipeline for transforming Text QA into Multimodal QA, and how to evaluate the quality of this transformation.

- The proposed pipeline and quality rubric are intuitively sensible.

- The authors have evaluated the transformation quality across multiple model families.

**Weaknesses:**

- The paper mentions establishing "a transformation pipeline for effective SMQ-to-MMQA transformation," yet it does not actually generate a large-scale MMQA dataset to train VLMs, which would be the definitive test of the data's quality. In other words, does an improved score on the MMQA generation benchmark actually correlate with higher data quality?
    - I believe the most reliable method for assessing data quality is to train a VLM on the synthetic data. The model's performance on downstream benchmarks would then reflect the true data quality, and one could verify if these results align with the rankings presented in Table 4.

- The paper should include a cost analysis of the Q-Mirror Agent. The Q-Mirror process involves multiple inference and refinement steps. If the resulting performance gains come at a computational cost that far exceeds the GPT family, it will likely be difficult to scale in practical applications.

**Questions:**

Why are both Line 382 and Line 389 labeled "Judge Alignment on Q-Mirror-Expert"?

**Details Of Ethics Concerns:**

nan

---

> ### Author Response · Authors · 2025-11-21
> **Official Responses to Reviewer D1fp**
>
> **1. Training VLMs on synthetic data**
>
> Thank you for your valuable feedback.
>
> First, we would like to emphasize that the core novelty of our work is to treat the TQA-to-MMQA transformation as a systematic evaluation problem and to build a complete, principled framework around it. We (i) introduce a **TQA-to-MMQA framework grounded in a multi-dimensional quality rubric** that defines a successful transformation, and (ii) turn this rubric into **two complementary benchmarks** that provide a unified protocol for comparing generators and judges. On top of this, we design the **Q-Mirror agentic system**, which operationalizes the framework in a closed loop by using the rubric and judge ensemble for iterative refinement. Thus, our contribution is a **reusable, evaluation-oriented framework serving as a quality control layer** for MMQA construction: it makes the transformation pipeline **transparent, measurable, and extensible**, and is explicitly designed so that the resulting data quality is aligned with human expert criteria.
>
> For higher data quality, we take several steps to provide direct evidence through human prefenrence. Our quality assessment criteria align closely with human expert judgments (Krippendorff’s Alpha = 0.82), which suggests that the defined criteria have high discriminative power**.** Furthermore, through the use of the **Q-Mirror Agent**, which is our initial attempt to enhance data generation, we **improve the quality of the generated MMQA data significantly**, with clear improvements in logic, consistency, and scientific correctness. These improvements are reflected in expert evaluations, confirming that our approach can indeed elevate and enhance the data quality in a way that is consistent with human judgment.
>
> While we recognize that training VLMs on synthetic data and testing them on downstream tasks is an important next step, this process is highly resource-intensive and time-consuming, making it challenging to complete within the rebuttal period. However, we are committed to pursuing this direction as part of our future work. Specifically, we plan to expand the scale of the MMQA dataset using the Q-Mirror Agent, and subsequently conduct downstream fine-tuning experiments on VLMs. This will allow us to directly measure the impact of data quality on model performance and validate the effectiveness of our quality evaluation framework.
>
> Once again, we sincerely appreciate the reviewer’s insightful comments. They have provided a clear direction for our future work, and we will continue to explore this important line of research moving forward.
>
>
>
> **2. Q-Mirror is scale in practical applications**
>
> Thank you for your valuable feedback. We acknowledge the importance of evaluating the computational cost of the Q-Mirror Agent, given its multi-step inference and refinement process.
>
> We estimate that the average cost per MMQA generation using Q-Mirror is **US $1.0–$2.0**, depending on the specific API pricing and settings (K=4 candidates, M=3 evaluators). For comparison, manual generation and validation of high-quality MMQA pairs by domain experts typically costs **several hundred dollars per item**, considering the time and labor required to create, diagram, and validate the content [1, 2].
>
> While SOTA models like GPT-4 are cheaper per inference (about US $0.2–$0.5 per item), the **pass rate** is much lower. Our experiments show that **GPT-4’s pass rate is only 72%**, meaning that around 28% of generated items fail to meet expert quality standards and require rework. In contrast, Q-Mirror raises the pass rate to **95%**, significantly reducing the waste of computational resources and manual intervention.
>
> The **additional computational cost** incurred by Q-Mirror should be seen as an investment in automated quality contro**l**, which ensures that the data generated is of high quality and minimizes the downstream costs associated with verification or re-generation. This makes Q-Mirror a more cost-effective solution for large-scale MMQA data generation than relying solely on manual methods or simple single-pass generation models.
>
> Further, we think that the appropriate comparison should focus on the **total cost of obtaining a verified high-quality MMQA pair**, rather than just per-inference costs. Manual generation can cost **several hundred dollars** and take **20–30 hours** of expert labor per item, whereas Q-Mirror achieves a much higher efficiency, producing high-quality data at a significantly lower total cost.
>
> We will also make sure to add this **cost analysis** to the **appendix** of the paper, providing the detailed calculations and further validation of our findings.

---

> ### Author Response · Authors · 2025-11-21
> **Official Responses to Reviewer D1fp**
>
> ###### Cost Comparison Table
>
> | Method                | Cost per Generation Attempt (USD) | Quality Pass Rate    |
> | --------------------- | --------------------------------- | -------------------- |
> | Manual Generation     | $200–$300                         | High                 |
> | SOTA Model Generation | $0.2–$0.5                         | Low (72% pass rate)  |
> | Q‑Mirror Generation   | $1.0–$2.0                         | High (95% pass rate) |
>
>
>
>
>
>
> **3. Correct typo**
>
> Thank you for your careful review and for pointing out the typo in our figure labels. We have made the necessary correction in the revised manuscript.
>
>
>
> References:
>
> [1] Griffin, B et al. *Auto‑Labeling Data for Object Detection*. arXiv:2506.02359.
>
> [2] iMerit. *Data Labeling for Machine Learning (All You Need to Know)*. Blog. 2021

---

### Official Review · Reviewer_MiWP · 2025-11-01

**Soundness:** 3
**Presentation:** 3
**Contribution:** 2
**Rating:** 4
**Confidence:** 4

**Summary:**

This paper presents a well-executed and clearly written framework for converting text-only scientific QA pairs into multimodal QA tasks. The methodology is systematic and reproducible, with clearly defined rubrics (IC, CM, QT) and thorough experimental validation. The proposed evaluation loop combining generation and judgment is well motivated and demonstrates consistent gains.

**Strengths:**

1. The paper is clearly written and well-structured, presenting a systematic framework for converting text-only scientific QA pairs into multimodal QA tasks.
2. The proposed evaluation rubric (IC, CM, QT) and the closed-loop generation-judgment process are well motivated and executed with solid experimental validation.
3. The methodology is practical, showing consistent gains and good alignment between automated and human evaluations.
4. Overall presentation quality is high, with organized sections, clear figures, and sound analysis.
5. The experiments are pretty complete and include a lot of open-sourced and closed-sourced models.

**Weaknesses:**

I would like to first say that I actually like the idea and the paper overall; it is a creative attempt to bridge text-only QA with multimodal reasoning. However, I do have one strong concern regarding the quality of the generated images.

In Figure 7, which the authors describe as a "successful example," the generated table headers are nearly illegible. Most characters are incomplete or incorrectly rendered. This kind of error is, to some extent, expected, as text-to-image models frequently struggle with accurate text transcription.

Similarly, in Figure 10, another claimed "successful case," the original question explicitly mentions that "An electron is moving along the horizontal axis from the left with a Lorentz factor of 4." This crucial information is completely missing in the corresponding figure. It is unclear how such a model could truly support reasoning if the generated visualization is missing essential quantitative context. Interestingly, this information appeared in an earlier version of the figure (see figure 11), but was removed in the later ones, which raises questions about the consistency and reliability of the generation process.

**Questions:**

As mentioned in the weaknesses, could the authors provide a few factually correct examples of generated images? Based on the current samples shown in the paper, it is difficult for readers to believe that the method truly works as intended, since even the "successful" figures contain clear factual or textual errors.

I would be happy to raise my score if the authors can demonstrate that the image generation process can reliably produce scientifically accurate and semantically consistent visualizations. Overall, my main concern lies in the factual correctness of the generated images rather than the overall idea, which I find promising.

---

> ### Author Response · Authors · 2025-11-21
> **Official Responses to Reviewer MiWP**
>
> We appreciate your positive feedback on the paper's concept and your rigorous scrutiny of the visual outputs.
>
> To address your concerns and demonstrate the reliability of our **Q-Mirror Agent**, we have added **Appendix F** with two high-quality case studies:
>
> **1）Data Table Legibility (Figure 12 & 13):** An astronomy case featuring a **perfectly legible data table** with complex values. Figure 13 further illustrates how the agent detects and corrects factual errors through its iterative loop, directly addressing your concern about reliability.
>
> **2）Logical & Factual Accuracy (Figure 14 & 15):** A biology case where a complex ecosystem description is transformed into a **factually accurate food web**, preserving 100% of the logic from the text.
>
> We believe these examples provide concrete evidence and respectfully invite you to review these new results in **Appendix F**.

---

> ### Comment · Reviewer_MiWP · 2025-11-27
>
> Thank you for the detailed response and for including the new examples in Appendix F&G. I appreciate the authors' effort to directly address my concerns about the factual quality of the generated images.
>
> The additional figures (especially Figures 12–15) indeed demonstrate noticeable improvement: the tables are now legible, and the visualizations generally preserve the logical and factual content of the text. Although a few minor textual errors remain (e.g., occasional spelling issues such as "planet" → "plant"), the overall quality is within an acceptable range and sufficient to support the paper’s main claims.
>
> I am glad to see these successful cases and that the authors took the feedback seriously. As promised, I will raise my score to 6. Good luck!

---

> > ### Author Response · Authors · 2025-11-27
> > **Official Responses to Reviewer MiWP**
> >
> > We are grateful for your positive feedback. We specifically appreciate your insightful suggestions regarding image quality, which significantly improved the quality of our paper.

---

### Meta-Review · Area_Chair_vUmp · 2026-01-07

**Summary:**

This paper studies the task of transforming text-only scientific QA pairs (TQAs) into multimodal QA pairs (MMQAs). It proposes a transformation pipeline, introduces a rubric to evaluate MMQA quality, benchmarks multiple model families on MMQA generation and evaluation, and analyzes alignment between automatic judges and human experts. The paper further proposes an agentic system (Q-Mirror) that iteratively refines MMQAs in a closed loop to improve generation quality.

Reviewer concerns concentrated on these themes:
- Reliability and factual quality of generated images. A major concern is whether the proposed models/pipelines can reliably generate accurate and faithful visuals. Reviewer MiWP highlighted illegible tables and missing critical quantitative information even in successful examples. Reviewers Y9jp and nagL also noted weak performance on Scientific Plausibility (SC).
- Evidence of utility and scalability. Reviewers questioned whether improvements on the proposed rubric scores translate to meaningful downstream value, and whether the approach is practically scalable. Reviewer D1fp emphasized that training a VLM on the synthetic MMQAs and evaluating on downstream benchmarks would be the most definitive validation of data quality.
- Validity and robustness of LLM-as-a-judge evaluation. Several reviewers raised concerns about self-confirmation bias and whether judge–human alignment is strong enough for the closed-loop refinement to be trustworthy.
- Benchmark size, representativeness, and metric design choices. Reviewers questioned whether the benchmark is sufficiently large and representative for the claims, and whether rubric dimensions/weights are theoretically justified and robust.

In rebuttal, the authors clarified that Q-Mirror is fully automatic at inference time, that human experts are used to validate the evaluation benchmarks rather than being part of the transformation pipeline, and that a judge ensemble improves alignment with human ratings. They also provided additional analyses to justify aspects of the rubric and weighting.

However, the most critical concerns remain only partially resolved: (i) the reliability of generated images, and (ii) the true downstream utility of the proposed task and rubric. In particular, both the task formulation and the evaluation rubric are introduced by the paper itself, but the submission lacks strong evidence that TQA-to-MMQA transformation yields data that is demonstrably useful for real-world applications, such as effective synthetic training data for VLMs, or that model performance under this rubric correlates with performance on established, well-annotated MMQA benchmarks. Strengthening this link to downstream validity would be important to fully justify the paper’s broader claims.

**Reviewer Concerns:**

Addressed concerns:
- Validity and robustness of LLM-as-a-judge evaluation: The authors clarified that Q-Mirror uses a fully automatic judge ensemble at inference time, provided evidence showing improved alignment between the ensemble and human experts.
- Metric design choices and weighting: Questions about the definition, orthogonality, and weighting of the rubric were addressed.
- Cost and scalability claims: A cost analysis was addressed with a quantitative comparison between Q-Mirror, single-pass generation, and manual expert curation.

Outstanding concerns:
- Reliability of generated images: Although the rebuttal included new illustrative examples, it remains unclear how consistently the pipeline produces scientifically accurate visuals at scale. While Q-Mirror improves Scientific Plausibility (SC) scores relative to baseline models, it is still uncertain whether the generated images are sufficiently reliable for real-world use cases.
- Downstream utility of the task and generated data: The authors argue that higher rubric scores indicate better MMQA quality, but there is no direct evidence that TQA-to-MMQA transformation produces data that is useful for downstream applications, such as training or improving VLMs.
- Benchmark size and representativeness: Concerns remain regarding the relatively small benchmark size and whether it is representative of real-world multimodal scientific reasoning tasks.

**Reviewer Scores:**

Reviewer MiWP explicitly indicated an increase in score. Reviewer D1fp is expected to keep their original score. For reviewers Y9jp and nagL, given the remaining concerns around Scientific Plausibility and lack of downstream evidence, I do not expect a score increase.

---

### Decision · Program_Chairs · 2026-01-26

Reject